# Mrj is a chaperone of the Hsp40 family that regulates Orb2 oligomerization and long-term memory in *Drosophila*

Meghal Desai[1☯], Hemant[1☯], Ankita Deo[2], Jagyanseni Naik[1], Prathamesh Dhamale[1], Avinash Kshirsagar[1], Tania Bose[2‡], Amitabha Majumdar[1]*

**1** National Centre for Cell Science, Savitribai Phule Pune University Campus, Pune, India, **2** Institute of Bioinformatics and Biotechnology (IBB), Savitribai Phule Pune University, Pune, India

☯ These authors contributed equally to this work.
‡ TB is co-senior author to this work.
* amitavamajumdar@gmail.com

**Data Availability Statement:** All data files are available from the figshare database (https://figshare.com/s/f5d913a0a289339ee16b).

## Abstract

Orb2 the *Drosophila* homolog of cytoplasmic polyadenylation element binding (CPEB) protein forms prion-like oligomers. These oligomers consist of Orb2A and Orb2B isoforms and their formation is dependent on the oligomerization of the Orb2A isoform. *Drosophila* with a mutation diminishing Orb2A's prion-like oligomerization forms long-term memory but fails to maintain it over time. Since this prion-like oligomerization of Orb2A plays a crucial role in the maintenance of memory, here, we aim to find what regulates this oligomerization. In an immunoprecipitation-based screen, we identify interactors of Orb2A in the Hsp40 and Hsp70 families of proteins. Among these, we find an Hsp40 family protein Mrj as a regulator of the conversion of Orb2A to its prion-like form. Mrj interacts with Hsp70 proteins and acts as a chaperone by interfering with the aggregation of pathogenic Huntingtin. Unlike its mammalian homolog, we find *Drosophila* Mrj is neither an essential gene nor causes any gross neurodevelopmental defect. We observe a loss of Mrj results in a reduction in Orb2 oligomers. Further, Mrj knockout exhibits a deficit in long-term memory and our observations suggest Mrj is needed in mushroom body neurons for the regulation of long-term memory. Our work implicates a chaperone Mrj in mechanisms of memory regulation through controlling the oligomerization of Orb2A and its association with the translating ribosomes.

## Introduction

Memory is the experience-dependent ability to preserve and recover information from the past. The molecular mechanism behind long-term memory is long-term potentiation (LTP). LTP is the persistent change observed in synaptic strength accompanied by structural changes like new synaptic growth and stabilization, caused due to repeated patterns of electrical stimulation [1–3]. Several studies ranging across species suggest that protein synthesis plays a crucial role in regulating long-term memory and LTP [4–15]. The translation regulator in Aplysia, cytoplasmic polyadenylation element binding (CPEB) protein is crucial for the maintenance

**Funding:** This work was supported by a Wellcome Trust-DBT India alliance grant (IA/I/13/2/501030), a DBT grant (BT/PR25893/GET/119/174/2017), CEFIPRA grant (6503-E) and NCCS intramural funding to AM. Work in TB lab was supported by a DBT Ramalingaswami fellowship (BT/RLF/Re-entry/54/2013) and an IYBA grant (BT/09/IYBA/2015/03). The funders had no role in study design, data collection and analysis, decision to publish, or preparation of the manuscript.

**Competing interests:** The authors have declared that no competing interests exist.

**Abbreviations:** CPEB, cytoplasmic polyadenylation element binding protein ; DLS, dynamic light scattering; ECM, extracellular matrix; LGMD, limb-girdle muscular dystrophy; LTP, long-term potentiation; NGS, normal goat serum; RAC, ribosome-associated chaperone; SDD-AGE, semi-denaturing detergent agarose gel electrophoresis.

phase of long-term facilitation and stabilization of learning-induced new synaptic growth [16,17]. Its *Drosophila* homolog Orb2 is necessary for the persistence of long-term memory and has among its mRNA targets, genes regulating protein turnover, synapse formation, and neuronal growth [18,19]. The mouse homologs CPEB1, CPEB2, and CPEB3 are also implicated in the regulation of memory processes [20–24].

Aplysia CPEB behaves like functional prion-like proteins [25–27]. Prions were discovered as protein-based infectious particles associated with neurodegenerative diseases like Creutzfeldt–Jakob disease, Scrapie, and Bovine spongiform encephalopathy [28]. For the disease-causing prions, they can exist in 2 distinct conformational variants: one monomeric and another oligomeric amyloid-like form. The oligomeric form is toxic, dominant, and self-perpetuating as it can convert the monomeric to the amyloid form. Many proteins sharing similar dominant and self-perpetuating properties were discovered in yeast and were classified as yeast prions [29–33]. In recent times, several proteins which are nontoxic and similar in characteristics to prions are discovered. In these, since the conversion to the oligomeric amyloid-like form is mediated by a signal and the amyloid form can have a beneficial physiological function, they are described as functional prions [34–38]. For Aplysia CPEB, it is suggested that synaptic stimulation causes CPEB to convert to its prion-like state and this state can self-sustain as long as monomers are getting synthesized. The prion-like oligomers can further regulate the protein synthesis of the target mRNAs needed for the maintenance of long-term memory. This model gets support from studies with *Drosophila* Orb2, where a point mutation in Orb2 that disrupted its prion-like oligomerization caused an impairment in the persistence of long-term memory [39,40]. Biochemically separated monomeric and oligomeric forms of Orb2 exhibit functional differences from each other. In in vitro translation assays with its target mRNAs, the monomer was found to act as a translational repressor by decreasing its poly-A tail length while the oligomer acted as a translation activator by elongating its poly-A tail [41]. This observation is similar to the idea that prions based on their conformational difference and oligomeric status can have different biochemical functions. These translation-enhancing oligomers when visualized using cryo-electron microscopy, appear as amyloids and like prions can seed the conversion of the monomers to the amyloid form [42].

Orb2 has 2 isoforms, Orb2A and Orb2B, both containing the common prion-like domain and a C terminal RNA-binding domain. The 2 isoforms differ in the position of the prion-like domain, enrichment in the brain, and propensity to aggregate. Orb2A has 8 amino acids while Orb2B has 162 amino acids, in front of the prion-like domain. In comparison to Orb2B, Orb2A is less abundant in the brain. Orb2A has a higher propensity to aggregate compared to Orb2B. Though both the Orb2A and Orb2B isoforms interact among themselves and are present in the oligomers, it is Orb2A that acts as a seed for Orb2 oligomerization. This is supported by several lines of evidence. Firstly, a deletion of Orb2A causes a drastic reduction in the formation of endogenous Orb2 oligomers in the brain. Secondly, a point mutation of the fifth Phenylalanine to Tyrosine (F5Y) in Orb2A reduces its ability to oligomerize and interferes with the maintenance of long-term memory [40]. This residue is specific to only Orb2A and is not present in Orb2B. Thirdly, genetic deletion experiments suggested Orb2A does not need its RNA-binding domain and Orb2B does not need its prion-like domain for the maintenance of memory [43]. Finally, the addition of the prion-like domain of Orb2A can seed monomeric Orb2B to oligomerize and result in its change to a translational activator [41]. All this evidence suggests oligomerization of Orb2A is crucial for the formation of Orb2 oligomers and the maintenance of memory. Hence in this study, we asked, what regulates the oligomerization of Orb2A, how this regulator affects the overall Orb2 oligomers in the brain, and if this regulator plays any role in long-term memory.

Taking a cue from the yeast prion literature, where the protein folding machinery/chaperones act as key regulators of prions, we hypothesize, chaperones may also play a role in the regulation of Orb2A oligomerization. In yeast, the Hsp70, Hsp40, and Hsp104 chaperones have been found to regulate the oligomerization and propagation of prions [29,31,44–46]. Using an immunoprecipitation-based screen and a yeast-based prion conversion assay, here we identify *Drosophila* Mrj as a regulator of Orb2A's prion-like conformational conversion. Mrj stands for the mammalian relative of DnaJ [47] and functions as a chaperone in mammals. Mammalian Mrj is now referred to as DNAJB6 according to the HUGO gene nomenclature [48]. We find *Drosophila* Mrj to behave similarly to mammalian DNAJB6 as a chaperone and interfere with the aggregation of pathogenic Huntingtin. While knockout of Mrj in mice is embryonic lethal [49], in *Drosophila* we find it not to be an essential gene and observe this knockout to have a reduced amount of Orb2 oligomers. We further find that Mrj knockout exhibits a deficit in long-term memory and Mrj is needed in specific mushroom body neurons for the regulation of long-term memory. Our observations suggest Mrj interacts with translating ribosomes and might play a role in regulating Orb2A's association with the translating ribosomes.

## Results

### Identification of chaperone interactors of Orb2

We started with identifying homologs of the yeast Hsp70, Hsp40, and Hsp104 families of proteins. The *Drosophila* genome lacks any Hsp104 homolog but it has members of the Hsp40 and Hsp70 classes of proteins [50–52]. So, for our screen to identify Orb2A regulators, we decided to focus on these 2 groups. The Hsp40 group of proteins is classified as proteins containing DnaJ domains. The *Drosophila* genome contains 39 such genes (S1 Fig). For the Hsp70 class, *Drosophila* has both the heat shock inducible Hsp70 (Hsp68, Hsp70Aa, Hsp70Ab, Hsp70Ba, Hsp70Bb, Hsp70Bc, and Hsp70Bbb) and the constitutively expressing Hsc70 (Hsc70-1, Hsc70-2, Hsc70-3, Hsc70-4, Hsc70-5, and Hsc70Cb) (S2 Fig). We made a library of 37 of the Hsp40 genes tagged with HA epitope and 4 of the Hsp70 genes tagged with Flag epitope. The 4 Hsp70 genes were selected based on their sequence variability from each other. We next transfected each of these constructs with Orb2A in *Drosophila* S2 cells and used these cells to perform immunoprecipitation with an anti-Orb2 antibody (Fig 1A and 1C). The immunoprecipitate was probed in a western blot for the presence of HA-tagged Hsp40 or Flag-tagged Hsp70 protein. From this immunoprecipitation-based screen, we observed the proteins CG4164, CG9828, DroJ2, CG7130, Tpr2, Mrj, Hsc70-1, Hsc70-4, Hsc70Cb, and Hsp70Aa as interactors of Orb2A (Fig 1B and 1D). For 31 Hsp40 proteins, we could not detect interaction with Orb2A in our immunoprecipitation screen (S3 Fig).

### *Drosophila* Mrj converts Orb2A from non-prion to a prion-like state

We next asked if any or all of these interactors can regulate the oligomerization of Orb2. Toward this, we tested these genes in a heterologous yeast chimeric Sup35-based system. Sup35 is a translation terminator that can exist in both non-prion and prion forms [29,53–55]. Replacement of Sup35's prion-like NM domain with putative prion-like domains of other proteins was previously used to identify several new prions [56,57]. In this assay, when the NM domain is replaced with the N terminal 162 amino acids of Orb2A (Orb2A-PrD), the prion-like behavior could be visualized as a red (non-prion) or white (prion) colony color. While the white colonies grow in adenine-deficient media, the red colonies are unable to grow in the same [39]. To check what effect the Orb2 interactors will have on the non-prion state of Orb2A, we made Galactose inducible constructs for expressing the 10 Orb2 interactors and transformed these into red-colored, adenine negative, Orb2A-PrD-C-Sup35 strain. The

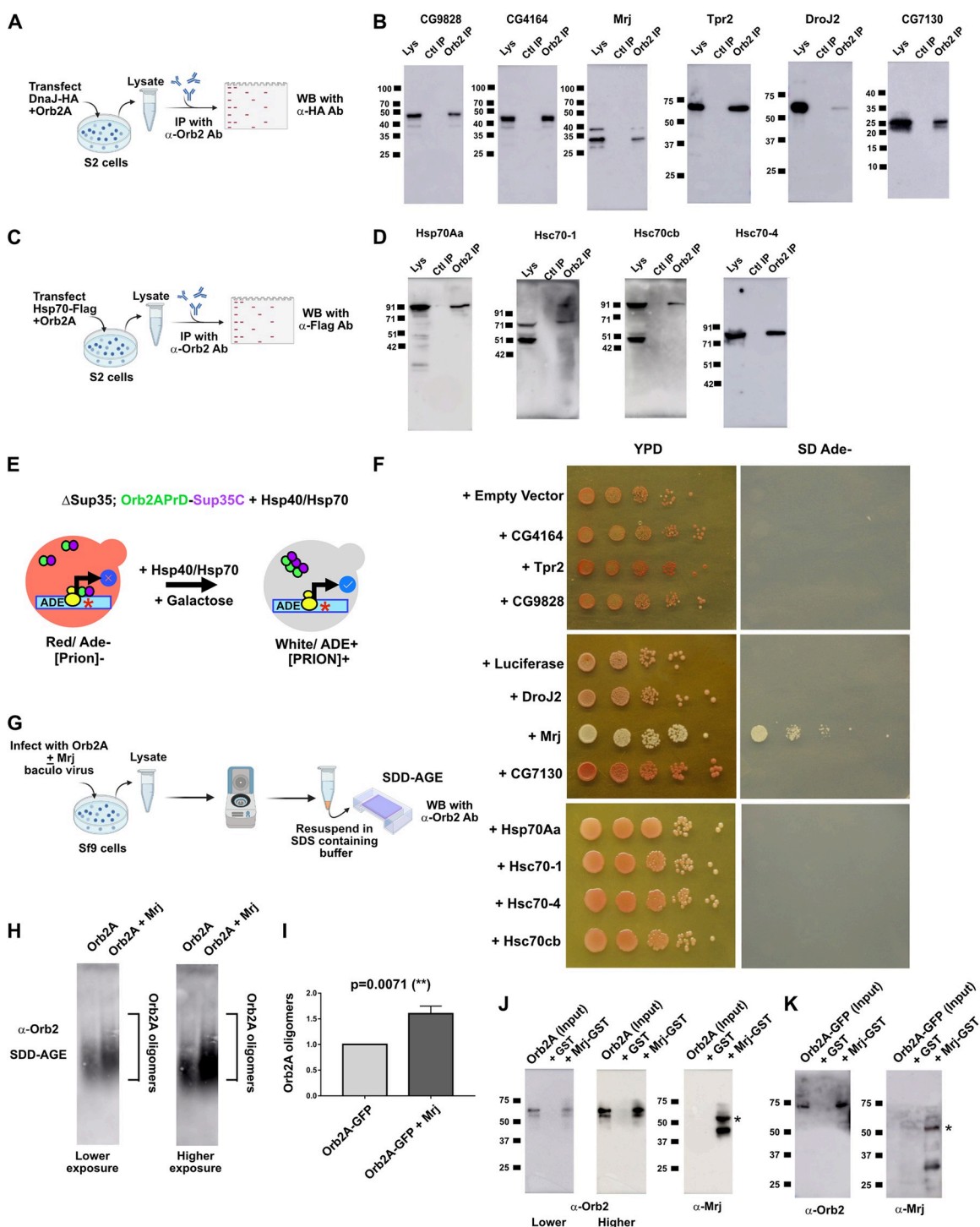

**Fig 1. Identification of Mrj as a regulator of Orb2A's prion-like oligomerization.** (**A**) Schematic of immunoprecipitation assay to identify Hsp40 interactors of Orb2A. Individual constructs coding for members of the Hsp40 family with HA tag were cotransfected in S2 cells with Orb2A construct and immunoprecipitation was performed with anti Orb2 antibody. The immunoprecipitate was probed with an anti-HA antibody to detect if the Hsp40 proteins were interacting with Orb2A. (**B**) From 37 proteins of the Hsp40 family, the immunoprecipitation (IP) screen identified CG9828, CG4164, Mrj, Tpr2, DroJ2, and CG7130 as interactors. Lys is the input lysate, Ctl IP is the control IP with beads only and Orb2 IP is the IP with anti-Orb2 antibody. (**C**) Schematic of immunoprecipitation assay to identify Hsp70 interactors of Orb2A. Individual constructs coding for members of the Hsp70 family with Flag tag were cotransfected in S2 cells with Orb2A construct and immunoprecipitation was performed with anti Orb2 antibody. The immunoprecipitate was probed

with an anti-Flag antibody to detect if the Hsp70 proteins were interacting with Orb2A. (**D**) All 4 proteins of the Hsp70 family screened in the immunoprecipitation experiment, Hsp70Aa, Hsc70-1, Hsc70Cb, and Hsc70-4 were found to be interacting with Orb2A. (**E**) Schematic of the yeast based screen. A Sup35 knockout strain rescued by a chimeric construct expressing Orb2A's prion-like domain tagged with the C-terminal domain of Sup35 is transformed with galactose inducible Hsp40 and Hsp70 constructs. A premature stop codon in the Ade 1–4 gene is used as a reporter. The screen consists of inducing the chaperones with galactose in the prion negative strain and screening for their ability to change the color to white in adenine-deficient media. (**F**) Galactose induction of individual chaperones of both Hsp40 and Hsp70 family of proteins in prion negative Orb2APrD-Sup35C strain, caused only Mrj to convert the prion negative state of the cells to prion positive state, as evidenced by the change in the colony color to white in YPD media and causing it to now grow in adenine. (**G**) Schematic of SDD-AGE assay to quantitate the change in Orb2A-GFP oligomerization in presence of Mrj. Sf9 cells were infected with viruses for Orb2A alone and Orb2A with Mrj. The lysate from these cells was centrifuged and the resulting pellet was resuspended in an SDS-containing buffer, subjected to SDD-AGE, and further probed with anti-Orb2 antibody. (**H**) Representative SDD-AGE blots showing increased levels of Orb2A oligomers in presence of Mrj. (**I**) Quantitation of Orb2A oligomers in presence and absence of Mrj. Data is represented as a relative fold change for Orb2A in presence of Mrj as compared to without Mrj. Data is represented as mean ± SEM and significance is tested using two-tailed Student's unpaired $t$ test. (**J**) Recombinant GST-Mrj bound to Glutathione beads on incubation with purified recombinant Orb2A-His from *E. coli* could pulldown Orb2A. The same blot is probed first with an anti-Orb2 antibody followed by probing with an anti-Mrj antibody. The band marked with * is the GST-Mrj protein and the band below it is possibly a breakdown product from the former. GST protein bound to Glutathione beads was used as a negative control. (**K**) Recombinant GST-Mrj bound to Glutathione beads on incubation with purified recombinant Orb2A-GFP-His from Sf9 cells could pulldown Orb2A-GFP. The same blot is probed first with an anti-Orb2 antibody followed by probing with an anti-Mrj antibody. The band marked with * is the GST-Mrj protein and the band below it is possibly a breakdown product from the former. GST protein bound to Glutathione beads was used as a negative control. The data underlying this figure are available at: https://figshare.com/s/f5d913a0a289339ee16b.

transformed colonies were independently grown and then induced with Galactose (Fig 1E) and plated with serial dilutions on complete media (YPD) and adenine-deficient media. Out of all the 10 genes from the Hsp40 and Hsp70 groups and controls, we observed only Mrj coexpression could change the red color of the colony into white. These cells could also now grow in adenine-deficient media (Fig 1F). This yeast-based screen of interactors of Orb2 suggested Mrj could convert the non-prion form of Orb2A to its prion-like state.

A conversion of Orb2A from its non-prion to prion-like form suggests an increase in its oligomeric state. We tested this in Sf9 cells by coexpressing Orb2A with and without Mrj. The cells were lysed and centrifuged and the pellet was further resuspended in an SDS-containing buffer and then resolved on a semi-denaturing detergent agarose gel electrophoresis (SDD-AGE) (Fig 1G) [56,58]. SDD-AGE has been previously reported to detect SDS-resistant Orb2 oligomers as a smear in agarose gels [40,41,59]. On probing the SDD-AGE blot with an anti-Orb2 antibody and quantitating the intensity of the smear, we observed a significant increase in Orb2 oligomers in the presence of Mrj (Fig 1H and 1I). This suggests that *Drosophila* Mrj can increase the oligomerization of Orb2A.

We further asked if Mrj can interact directly with Orb2A and to address this used recombinant protein-based pulldown assays using GST-Mrj bound to Glutathione beads. We added these beads to either recombinant Orb2A (6X Histidine tagged from *E. coli*) or recombinant Orb2A-GFP (6X His tagged from Sf9 cells) and on performing western blots with the pulldown beads noticed the presence of Orb2A in both these cases (Fig 1J and 1K). Overall, these pulldown experiments suggest *Drosophila* Mrj can directly interact with Orb2A.

### *Drosophila* Mrj like its mammalian homolog interacts with Hsp70 proteins and can act as a chaperone preventing aggregation of pathogenic Huntingtin

The amino acid sequence of *Drosophila* Mrj is well conserved with its homologs in humans, mouse, rat, frog, and zebrafish (S4A Fig). According to Flybase, Mrj has 3 possible translated products of 259, 346, and 208 amino acid lengths (S1 Data). On performing RT-PCR using RNA purified from fly heads, we were able to detect only the 259 amino acid fragment (S4B Fig) and this particular fragment is used in the previous immunoprecipitation screen, the

prion conversion assay, and in all subsequent experiments. We asked if *Drosophila* Mrj has similar properties to mammalian Mrj. For the sake of clarity from here onwards, we will refer to mammalian Mrj as DnaJB6 as per the HUGO gene nomenclature [48]. Similar to mammalian DNAJB6 [47], *Drosophila* Mrj is also present in both the cytoplasm and nucleus (S4C Fig). Mammalian DNAJB6 is reported to self-interact to form oligomers [60–63]. We tested the self-association of *Drosophila* Mrj in vivo by coexpressing Mrj-RFP and Mrj-HA in S2 cells and immunoprecipitating Mrj-RFP (Fig 2A). On probing the immunoprecipitate, we could detect the presence of Mrj-HA in it (Fig 2B), suggesting that *Drosophila* Mrj self-associates inside cells. We also observed purified recombinant Mrj precipitate within a few days of purification suggesting the formation of higher-order oligomers that eventually phase out of the solution. Using dynamic light scattering (DLS) of purified recombinant Mrj, we noticed different sized Mrj populations in solution, which over 3 days moved to higher size suggesting the self-association of the protein (S4D Fig).

The Hsp40 chaperones interact with the Hsp70 family of proteins and act as cofactors for different substrates [64]. Mammalian Mrj is reported to interact with Hsp70 proteins [47,65,66]. Co-immunoprecipitation experiment of *Drosophila* Mrj with 4 of the *Drosophila* Hsp70 proteins suggested *Drosophila* Mrj can interact with at least 2 of these proteins Hsp70Aa and Hsc70-4 (Fig 2C and 2D). All these 4 immunoprecipitation experiments were done under the same conditions. To further test if the interactions of Mrj with Hsp70Aa and Hsc70-4 are direct or not, we mutated the HPD motif of Mrj to QPD. The HPD motif is needed for the interaction of DnaJ proteins with Hsp70 family proteins. In our immunoprecipitation experiment, we could not detect any interaction of the mutated Mrj with Hsp70Aa and Hsc70-4 (Fig 2E and 2F), suggesting their interaction is direct and specific.

Mammalian DNAJB6 could inhibit the aggregation and toxicity of Huntingtin Exon1 protein containing 150 Q repeats [67]. *Drosophila* Mrj was found to colocalize with expanded poly Glutamine aggregates in the brain and could rescue its toxicity in the *Drosophila* eye [68]. We tested the effect of *Drosophila* Mrj on Htt oligomers and aggregates using biochemical and imaging-based assays. Towards this, we coexpressed an Htt construct spanning the exon1 with 103 Q repeats [69] and observed colocalization of Mrj-RFP with HttQ103 aggregates in S2 cells (Fig 2G). The observation of colocalization of Mrj-RFP with Htt aggregates is similar to previous reports of chaperones showing colocalization with the aggregates [70–72]. On coexpression of Mrj-HA as well as of Mrj-RFP with HttQ103-GFP, we observed a significant decrease in the number of cells showing visible aggregates in comparison to coexpression with a control DnaJ domain-containing protein CG7133 (Fig 2H and 2I). We did not see any detectable difference in total Htt levels in the lysates from cells coexpressing with CG7133 and Mrj in SDS-PAGE (Fig 2J and 2K). We next ran the same extract in SDD-AGE. The smear representing the Htt oligomers in the Mrj lane is much lighter and covers a smaller range than the CG7133 lane indicating fewer oligomers (Fig 2L and 2M). Here, in the presence of Mrj, we noticed 2 distinct bands of Htt. One faster-moving band of lighter intensity migrates faster in comparison to the lower-most band in the CG7133 lane. The other band in the Mrj lane is a bit higher but is more intense in the blots and comes at a slightly higher position (than the lowermost band seen in the CG7133 lane). This intense band probably represents the collapse of the larger Htt oligomers to smaller, less diverse-sized Htt oligomers in the presence of Mrj, suggesting Mrj possibly interferes or impedes HttQ103 aggregation. We also had a similar observation of Mrj decreasing Htt aggregate and oligomers (S4E–S4G Fig) for a longer Htt construct spanning the caspase cleaved 588 amino acid fragment [73,74]. Together the observations of *Drosophila* Mrj interacting with Hsp70 proteins and its ability to interfere with pathogenic Htt aggregate formation suggests its possible role as a chaperone.

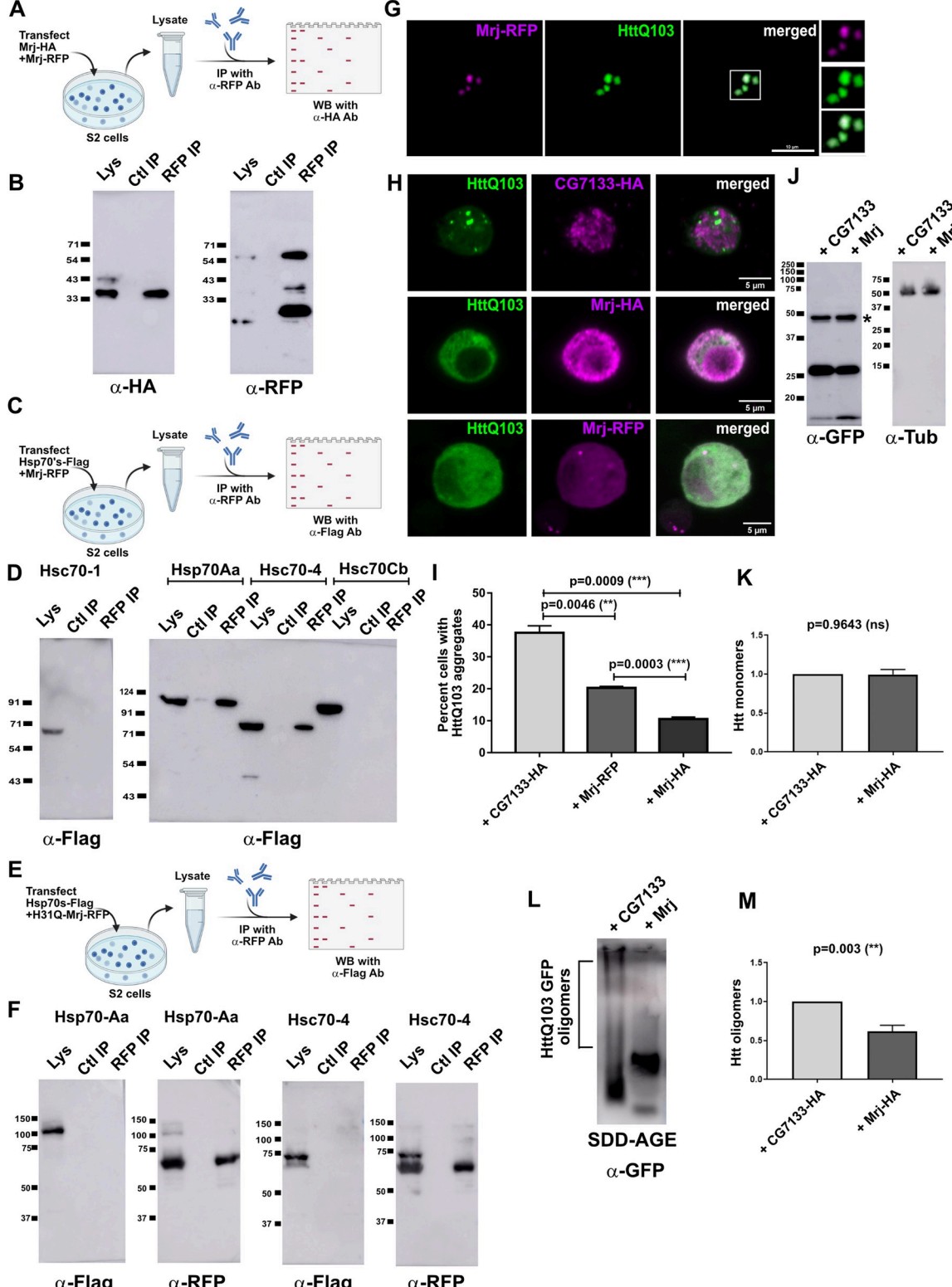

**Fig 2. *Drosophila* Mrj behaves like its mammalian homolog as a chaperone preventing Huntingtin aggregation.** (**A**) Schematic of immunoprecipitation assay to check the possibility of self-oligomerization of Mrj. Two Mrj constructs, Mrj-HA and Mrj-RFP were cotransfected in S2 cells. The cells were lysed and immunoprecipitation was done with anti-RFP (RFPTrap beads). The immunoprecipitate was next probed with anti-HA antibody. (**B**) Left panel shows a western blot of immunoprecipitated Mrj-RFP

probed with anti-HA antibody indicating the presence of Mrj-HA in the immunoprecipitate, suggesting an interaction between Mrj-RFP and Mrj-HA. The right panel shows the same blot probed with an anti-RFP antibody to confirm the pulldown of Mrj-RFP. (**C**) Schematic of immunoprecipitation assay to identify Hsp70 interactors of Mrj. Flag-tagged Hsp70 constructs were cotransfected with Mrj-RFP. Lysate from these cells was used for immunoprecipitation with anti-RFP beads, and the immunoprecipitate was probed with anti-Flag antibody. (**D**) In the IP experiment, of the 4 Hsp70's tested here, Hsp70Aa and Hsc70-4 show their presence in the immunoprecipitate suggesting of their interaction with Mrj. (**E**) Schematic of immunoprecipitation assay to check if Hsp70Aa and Hsc70-4 interacts with HPD motif mutated Mrj. Flag-tagged Hsp70 constructs were cotransfected with Mrj H31Q-RFP. Lysate from these cells was used for immunoprecipitation with anti-RFP beads, and the immunoprecipitate was probed with anti-Flag antibody. (**F**) In the IP experiment on probing with anti-Flag antibody both Hsp70Aa and Hsc70-4 could not be detected in the immunoprecipitate. The same blots when probed with an anti-RFP antibody, confirmed the pulldown of Mrj H31Q-RFP. (**G**) Representative confocal image of a single S2 cell coexpressing Mrj-RFP and Httexon1-Q103-GFP shows colocalization of Mrj with Htt aggregates. The scale bar is 10 microns. (**H**) Representative images of Httexon1Q103-GFP cells coexpressing with CG7133-HA, Mrj-RFP, and Mrj-HA suggest a decrease in the Htt aggregates in presence of Mrj. Scale bars are 5 microns. (**I**) Quantitation of the percentage of HttQ103-exon1GFP expressing cells with aggregates in presence of CG7133-HA, Mrj-RFP, and Mrj-HA suggests a significant decrease of Htt aggregates in presence of both Mrj-RFP and Mrj-HA. The Mrj-HA construct is more efficient in decreasing Htt aggregation compared to Mrj-RFP. Data is represented as mean ± SEM and significance is tested with two-tailed Student's paired $t$ test. (**J**) Western blot of lysates from S2 cells coexpressing Httexon1Q103-GFP with Mrj and CG7133 shows similar amounts of Htt (monomer size marked with *) in SDS-PAGE. Right panel shows the same lysates probed with anti-Tubulin antibody. (**K**) Quantitation of the Htt bands from SDS-PAGE shows no significant difference. Data is represented as mean ± SEM and significance is tested with two-tailed Student's unpaired $t$ test. (**L**) SDD-AGE from S2 cell lysate coexpressing Httexon1Q103-GFP along with CG7133 and Mrj showed a decreased amount of Htt oligomers in presence of Mrj. (**M**) Quantitation of SDD-AGE shows a significantly decreased levels of Htt oligomers in presence of Mrj. Data is represented as mean ± SEM and significance is tested with two-tailed Student's unpaired $t$ test. The data underlying this figure are available at: https://figshare.com/s/f5d913a0a289339ee16b.

## *Drosophila* Mrj knockout is viable and does not show any gross developmental defect

We next used the CRISPR-Cas9 system to generate a knockout of *Drosophila* Mrj by introducing a Gal4 cassette in the Mrj locus (Fig 3A). We confirmed the knockout by using both PCR and western blotting with an anti-Mrj antibody (Fig 3B and 3C). Here, we found the *Drosophila* Mrj knockout line to be homozygous viable. Developmentally, there was no defect in any organization of any body structure, and there was no sterility associated with the homozygous line. Using the Mrj KO Gal4 line with reporter CD8GFP, we could detect strong expression in the brain including areas of optic lobes, olfactory lobes, and mushroom body (S5D Fig). In terms of overall cellular organization, we observed no defect in the actin cytoskeleton and the gross organization of the mushroom body as evidenced by staining with Phalloidin and anti-FasII antibody (Fig 3D). We next checked if the absence of Mrj caused endogenous proteins to aggregate in the brain. Towards this, we looked for the *Drosophila* homolog of p62, Ref(2)P which is a regulator of protein aggregates and is present in ubiquitinated protein aggregates associated with aging and neurodegenerative disorders [75]. On Ref(2)P immunostaining to label ubiquitinated protein aggregates, we could not detect any difference in the labeling between the wild-type and Mrj knockout brains (Fig 3E). We also stained the wild-type and Mrj knockout brains with anti-ubiquitin antibody and again saw no difference in ubiquitin labeling in the brains (Fig 3F).

As mutations in human DNAJB6 are associated with limb-girdle muscular dystrophy (LGMD), we checked if the muscles in Mrj knockout fly show any defect in terms of organization or degeneration. Using the Mrj KO Gal4 line with CD8GFP as a reporter, we could see the expression of the reporter GFP in the muscles and NMJ synapses (S5E–S5G Fig) suggesting the similarity of *Drosophila* Mrj with human DnaJB6 expression. On staining the larval muscles with phalloidin, we observed no evidence of degeneration or disorganization in the Mrj knockout compared to the wild type (Fig 3G). We next checked the adult Mrj knockout fly for any defect in locomotion and found them to show no significant difference in their walking speed in comparison to the wild type (Fig 3H). This ruled out the possibility of any drastic muscle defect later in development. In terms of life span, we observed no difference between

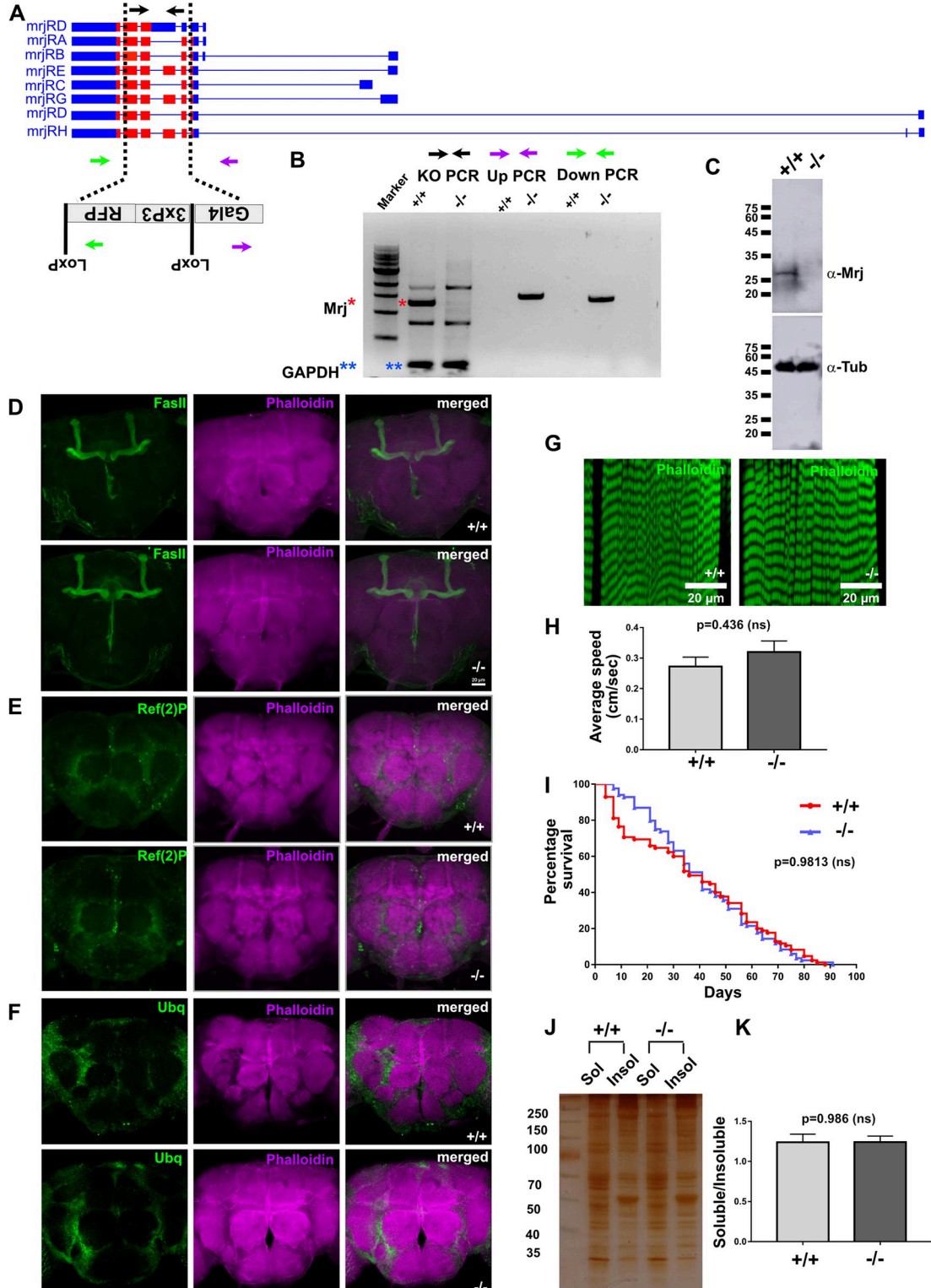

**Fig 3. *Drosophila* Mrj is not an essential gene unlike its mammalian homolog.** (**A**) Schematic of the genomic organization of the Mrj gene and the knockin of the Gal4-loxP-3XP3-RFP-loxP cassette in the locus to make Mrj knockout. The black arrows represent the PCR primers for confirming the Mrj knockout, and the green and purple arrows represent the PCR primers for up and down PCR to check the knockin of the cassette in the Mrj locus. (**B**) Confirmation of Mrj knockout using genomic DNA PCR with the knockout, up and down PCR primers. The red * marks the amplified band for Mrj in the wild type (+/+) and its absence

in the Mrj knockout (-/-). GAPDH PCR amplification (blue **) was used as the loading control in the PCR. (**C**) Confirmation of Mrj knockout using western blot using an anti-Mrj antibody. Anti-α-Tubulin antibody was used as the loading control. (**D**) Immunostaining of wild type (+/+) and Mrj knockout (-/-) with anti-FasII antibody and Phalloidin shows no gross difference in the overall morphology of the mushroom body and the brain. Scale bar is 20 micron. (**E**) Immunostaining of wild type (+/+) and Mrj knockout (-/-) with anti-Ref(2)P antibody and Phalloidin and (**F**) with anti-Ubiquitin antibody and Phalloidin shows no gross difference between the two sets. (**G**) Phalloidin staining of muscles from third instar larvae of wild type (+/+) and Mrj knockout (-/-) shows no gross difference. (**H**) Quantitation of the average speed of wild type (+/+) and Mrj knockout (-/-) shows no significant difference between the two. Data is represented as mean ± SEM and Mann–Whitney U test is used to check significance. (**I**) Kaplan–Meier survivor curve shows no significant difference in the life span of wild type (Red) and Mrj knockout (-/-) (Blue). Significance was tested with log-rank Mantel–Cox test. (**J**) Representative image of silver-stained gel of soluble and insoluble protein fractions, from wild type (+/+) and Mrj knockout (-/-) fly heads. (**K**) Quantitation of soluble to insoluble protein ratio from wild-type (+/+) and Mrj knockout (-/-) flies showed no significant difference. Data is represented as mean ± SEM and a two-tailed Student's unpaired *t* test is used to check significance. The data underlying this figure are available at: https://figshare.com/s/f5d913a0a289339ee16b.

the knockout and wild-type animals (Fig 3I). Since Mrj is known to prevent the formation of pathogenic Htt aggregates, we next asked if Mrj knockout animals show any difference in the total protein distribution in the brain between the soluble and insoluble fractions in comparison to the wild type. We lysed fly heads in Triton X-100 containing lysis buffer and separated the supernatant as the soluble fraction. The remaining pellet was solubilized using an SDS containing lysis buffer and separated as an insoluble fraction. We resolved these fractions in SDS-PAGE, performed silver staining (Fig 3J), and quantitated the ratio of soluble/insoluble protein. Here, we also observed no significant difference in terms of the amount of soluble/insoluble protein ratio between the wild-type and Mrj knockout animals (Fig 3K). These results together suggested that under normal growing conditions, Mrj knockout animals were indistinguishable from wild-type ones in terms of their development, brain and muscle organization, and locomotion abilities. Also, for ubiquitinated aggregates and insoluble protein amounts no difference could be seen between the two sets.

### *Drosophila* Mrj interacts with both the Orb2A and Orb2B isoforms independent of their RNA-binding domain

On co-transfecting Mrj-RFP with Orb2A-GFP in S2 cells, we could observe colocalization between Orb2A punctae and Mrj (Figs 4A, S5B and S5C). On immunoprecipitation of Mrj, we were able to detect the presence of Orb2A-GFP in the immunoprecipitate (Fig 4G). To address which region of Orb2A is needed for its interaction and colocalization with Mrj, we cotransfected different deletion mutants of Orb2A with Mrj-RFP and performed imaging and co-immunoprecipitation experiments (Fig 4F). Upon co-transfecting the Orb2A325-GFP construct which lacks the C terminal RNA-binding domain with Mrj RFP, we observed colocalization between the two (Figs 4B, S5B and S5C). Immunoprecipitation of Mrj from these cells could pull down Orb2A325 (Fig 4H). This suggested the interaction between Mrj and Orb2A is independent of its RNA-binding domain. We also cotransfected Mrj with an Orb2AΔ162-GFP construct which lacks the prion-like domain. On imaging, we could not detect any colocalization with Mrj (Figs 4C, S5B and S5C), and in immunoprecipitation experiments, observed the inability of Mrj to pull down Orb2AΔ162 (Fig 4I). This suggested the interaction of Mrj with Orb2A to be dependent on the N terminal 162 amino acid long region of Orb2A containing the prion-like domain. This also supported the previous observation of Mrj converting the Orb2A-PrD-Sup35 from a non-prion to a prion-like state, where the N terminal 162 amino acids of Orb2A were used to make the chimera with Sup35.

We also looked at the other Orb2 isoform Orb2B. Though Orb2B-GFP expressing cells rarely show punctate appearance characteristic of aggregation, when formed, these punctae showed colocalization with Mrj-RFP (Figs 4D, S5B and S5C). In immunoprecipitation

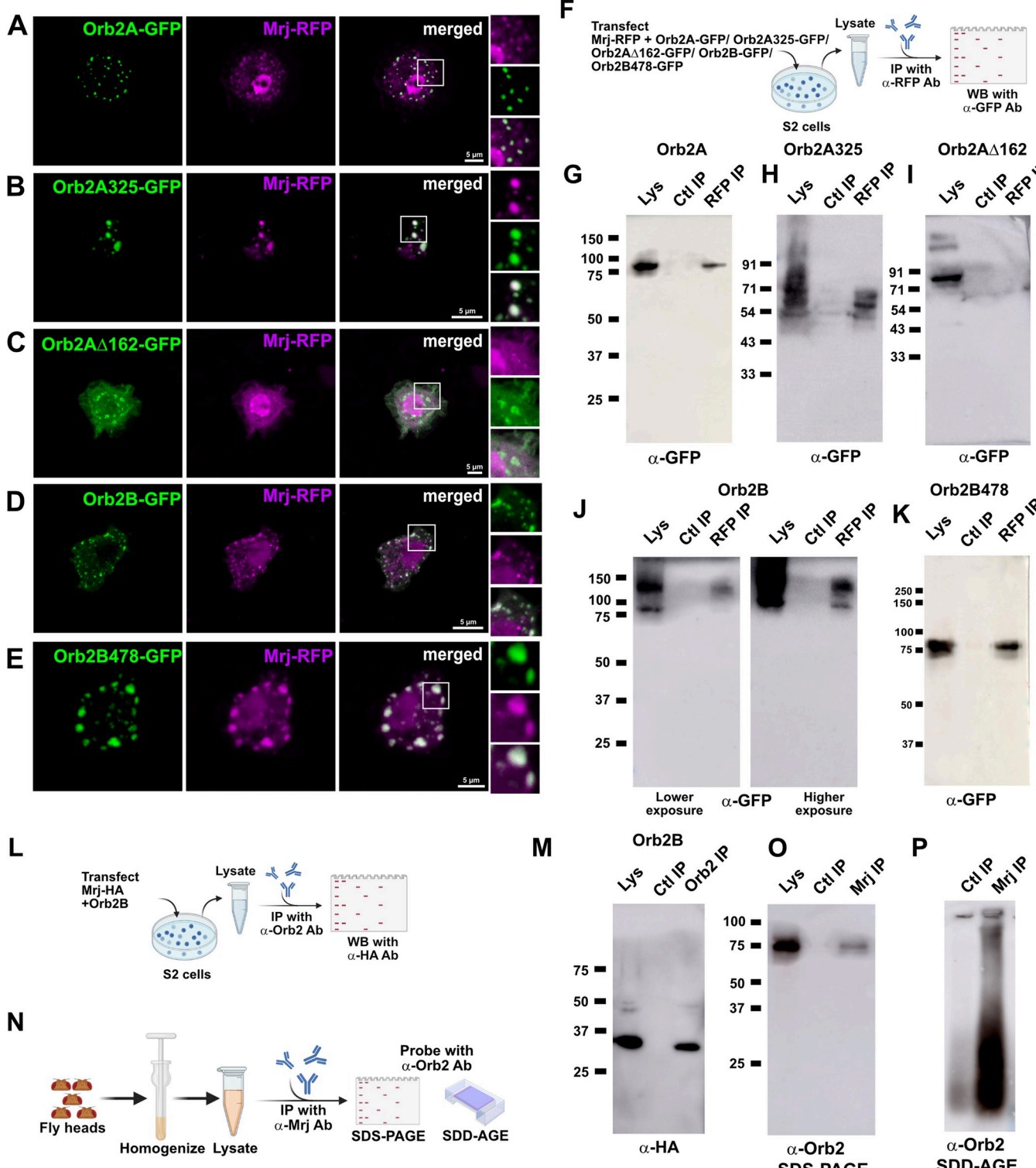

**Fig 4. Mrj interacts with both the Orb2 isoforms.** (**A**) Representative image of S2 cell co-expressing Orb2A-GFP and Mrj-RFP showing colocalization of Mrj with Orb2A aggregates. (**B**) Representative image of S2 cell co-expressing Orb2A325-GFP (RNA-binding domain deleted construct) and Mrj-RFP showing colocalization of Mrj with Orb2A325 aggregates. (**C**) Representative image of S2 cell co-expressing Orb2AΔ162-GFP (prion-like domain deleted construct) and Mrj-RFP showing no colocalization of Mrj with Orb2AΔ162. (**D**) Representative image of S2 cell co-expressing Orb2B-GFP and Mrj-RFP showing colocalization of Mrj with Orb2B aggregates. (**E**) Representative image of S2 cell co-expressing Orb2B478-GFP and Mrj-RFP showing colocalization of Mrj with Orb2B478 aggregates. (**F**) Schematic of immunoprecipitation assay to check for interaction between Mrj-RFP and Orb2A-GFP, Orb2A325-GFP,

Orb2AΔ162-GFP, Orb2B-GFP, and Orb2B478-GFP. (**G**) Immunoprecipitated Mrj-RFP pulls down Orb2A-GFP. (**H**) Immunoprecipitatated Mrj-RFP pulls down Orb2A325 suggesting the interaction between Mrj and Orb2A is independent of its RNA-binding domain. (**I**) No interaction detected between Orb2AΔ162-GFP and Mrj-RFP in immunoprecipitation experiment suggesting the interaction of Orb2A with Mrj is dependent on the prion-like domain of Orb2A. (**J**) Immunoprecipitated Mrj-RFP pulls down Orb2B-GFP. (**K**) Immunoprecipitated Mrj-RFP pulls down Orb2B478-GFP suggesting the interaction of Mrj with Orb2B is independent of its RNA-binding domain. (**L**) Schematic of immunoprecipitation assay to check for interaction between Mrj-HA and Orb2B. (**M**) Immunoprecipitation experiment showed interaction between Orb2B and Mrj-HA. (**N**) Schematic of immunoprecipitation assay to check for interaction between endogenous Mrj and Orb2. (**O**) Probing the SDS-PAGE of Mrj immunoprecipitate with anti-Orb2 antibody detects the presence of monomeric Orb2B. (**P**) Probing the SDD-AGE of Mrj immunoprecipitate with anti Orb2 antibody shows the presence of Orb2 oligomers. The data underlying this figure are available at: https://figshare.com/s/f5d913a0a289339ee16b.

experiments, on pulling down Mrj, we detected Orb2B in the immunoprecipitate (Fig 4J). We also performed immunoprecipitation from cells coexpressing Mrj-HA with untagged Orb2B (Fig 4L), similar to our earlier experiment of immunoprecipitation from cells coexpressing Mrj-HA with untagged Orb2A. Here, on pulling down Orb2B, we were able to detect Mrj-HA in the immunoprecipitate (Fig 4M). We also made an RNA-binding domain deletion of Orb2B (Orb2B478-GFP) and this showed colocalization with Mrj-RFP (Figs 4E, S5B and S5C). In the immunoprecipitation experiment, we were able to detect an interaction of Orb2B478-GFP with Mrj-RFP (Fig 4K). Overall, these experiments suggest Mrj interacts with both the Orb2A and Orb2B isoforms, and this interaction is independent of their RNA-binding domain.

As the endogenous Orb2 oligomers in the *Drosophila* brain consist of both Orb2A and Orb2B, we asked if endogenous Mrj interacts with endogenous Orb2 in the brain and if it interacts with the monomer or the oligomer or with both. We pulled down endogenous Mrj using anti-Mrj antibody and ran the immunoprecipitate in both SDS-PAGE and SDD-AGE followed by probing them with anti-Orb2 antibody (Fig 4N). Here, we noticed the presence of both monomeric Orb2B and oligomeric Orb2 in the immunoprecipitate (Fig 4O and 4P), suggesting endogenous Mrj interacts with both forms.

### *Drosophila* Mrj knockout shows reduced amounts of Orb2 oligomers

We next asked what happens to endogenous Orb2 oligomers in Mrj knockout animals. In western blots from fly heads, the detectable form is Orb2B representing its relatively higher abundance in the brain. We could not detect any significant difference in Orb2B levels between the wild type and Mrj knockout (Fig 5A). Next, we separated the soluble and insoluble fractions from wild-type and Mrj knockout fly heads and probed them with an anti-Orb2 antibody (Fig 5B). In this assay, while we could detect the presence of Orb2B in both the soluble and insoluble fractions in the wild-type fly head lysate, in the Mrj knockout, Orb2B was more enriched in the soluble fraction (Fig 5C and 5D). This indicated that Mrj knockout has less Orb2B in the insoluble fraction, indicating lighter or lesser Orb2 oligomers. We further tested this by immunoprecipitating Orb2 from wild-type and Mrj knockout animals and subjecting the immunoprecipitate to SDD-AGE (Fig 5E). On probing the SDD-AGE blots, we observed the presence of significantly reduced levels of Orb2 oligomers in Mrj knockout brains in comparison to the wild type (Fig 5F and 5G). Since the insoluble fraction here was solubilized using 0.1% SDS containing buffer, we wondered if the reduced amounts of Orb2 oligomers that we detect in the absence of Mrj are actually due to a reduction in Orb2 oligomerization or due to the migration of Orb2 oligomers to even more aggregated state which cannot be solubilized using 0.1% SDS. To address this, we took the wild-type and Mrj knockout fly heads, lysed them first in the TritonX-100 containing solubilizing buffer, the pellet was further solubilized using 0.1% SDS containing buffer. The remaining pellet was again resolubilized using 2% SDS containing buffer (Fig 5H). On running these different fractions in SDD-AGE we observed while there is more Orb2 present in the Triton X-100 soluble fraction from Mrj knockout, the

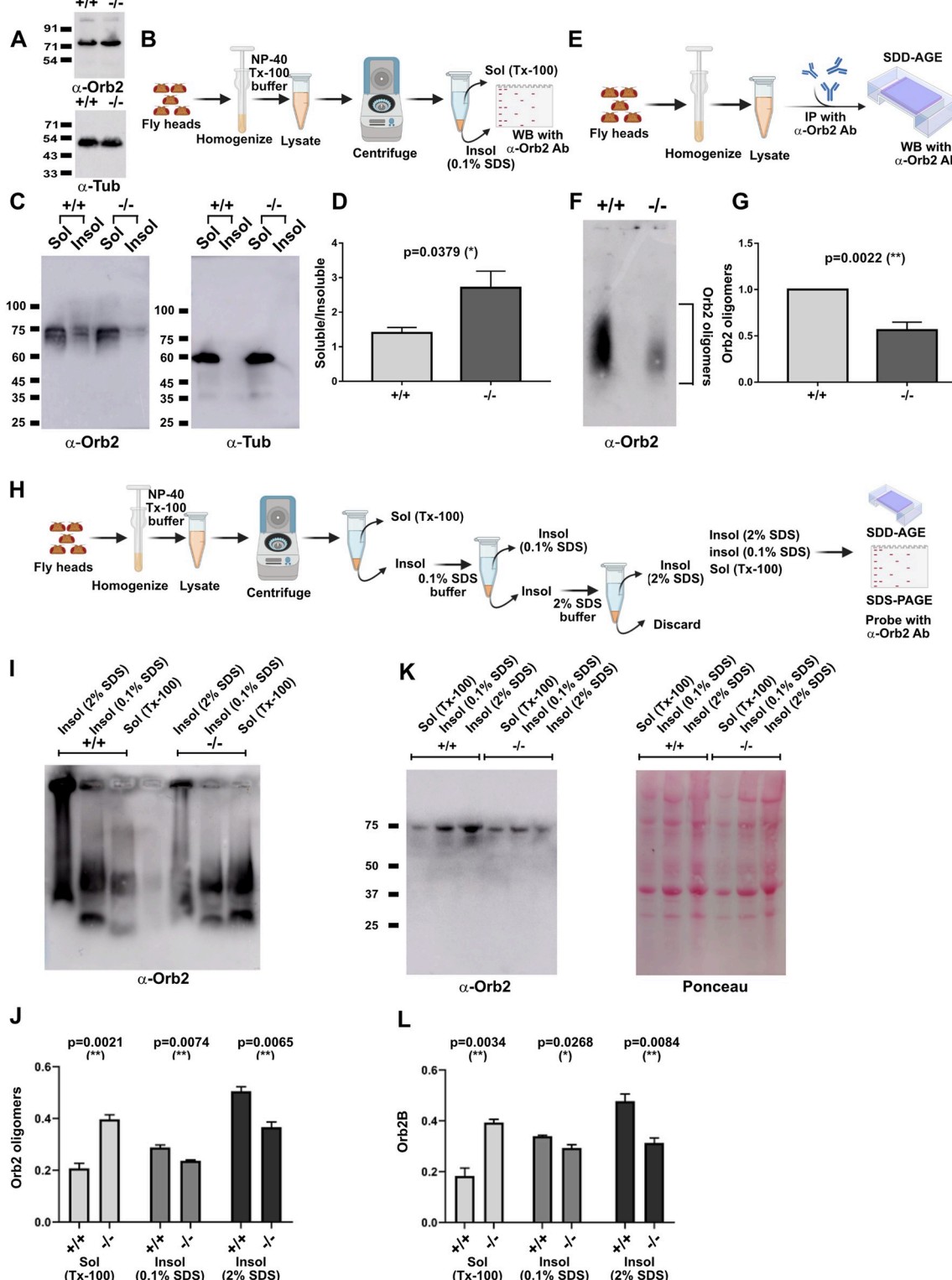

**Fig 5. Mrj knockout shows reduced Orb2 oligomers.** (**A**) Western blot of fly head extracts from wild type and Mrj KO shows similar amounts of Orb2B in both of them. α-Tubulin is used as a loading control here. (**B**) Schematic of the process of preparing the soluble and insoluble fractions from fly head extract to check for differential distribution of Orb2B. (**C**) Western blot of soluble and insoluble fractions from wild-type and Mrj KO fly heads show a reduced amount of Orb2B in the insoluble fraction of Mrj knockout. α-Tubulin was used as the loading control. (**D**) Quantitation of relative Orb2B level ratio in soluble to insoluble fractions show a significant

increase in Mrj KO flies as compared to control wild-type flies. Data is represented as mean ± SEM and significance is tested using two-tailed Student's paired $t$ test. (**E**) Schematic of the Orb2 pulldown experiment followed by SDD-AGE to check for Orb2 oligomers. (**F**) Representative SDD-AGE blot shows the reduced amount of Orb2 oligomers in Mrj knockout. (**G**) Quantitation of Orb2 oligomers in Mrj knockout and wild type. Data is represented as mean ± SEM and two-tailed Student's unpaired $t$ test is used to check significance. (**H**) Schematic of sequential detergent-based extraction using TritonX-100, 0.1% SDS, and 2% SDS containing buffers to check for differential distribution of Orb2 in wild type and Mrj knockout. (**I**) Probing the SDD-AGE of different fractions from wild type and Mrj knockout fly heads show, an increased presence of Orb2 in the TritonX-100 extracted fraction from Mrj knockout in comparison to the wild type. However, for both the 0.1% and 2% SDS extracted fractions the Orb2 oligomer levels were found to be reduced in the Mrj knockout. (**J**) Quantitation of the SDD-AGE shows increased levels of Orb2 oligomer in the TritonX-100 soluble fraction in Mrj knockout and decreased levels in the 0.1% and 2% SDS soluble fractions. Data is represented as mean ± SEM and two-tailed Student's unpaired $t$ test is used to check significance. (**K**) Probing the SDS-PAGE of the same extracted fractions with anti-Orb2 antibody shows a similar trend of increased amounts of Orb2B monomers in the TritonX-100 extracted fraction and decreased amounts of Orb2B monomers in the 0.1% and 2% SDS extracted fractions from Mrj knockout in comparison to wild type. For the 2% SDS extracted fraction, 25% of the total eluate was run to avoid saturation of the signal. The right panel shows the same blot stained with Ponceau stain. (**L**) Quantitation of the SDS-PAGE shows increased levels of Orb2B in the TritonX-100 soluble fraction in Mrj knockout and decreased levels in the 0.1% and 2% SDS soluble fractions. Data is represented as mean ± SEM and two-tailed Student's unpaired $t$ test is used to check significance. The data underlying this figure are available at: https://figshare.com/s/f5d913a0a289339ee16b.

amount present in both the 0.1% and 2% SDS soluble fractions was decreased in comparison to the wild type (Fig 5I and 5J). SDS-PAGE and immunoblotting with anti-Orb2 antibody showed a similar trend for the monomeric Orb2B (Fig 5K and 5L). So, overall these results suggest that the Mrj knockout has decreased amounts of Orb2 oligomers and thus the chaperone Mrj is probably regulating the oligomeric status of Orb2. An alternate possibility is early interactions of Mrj with Orb2 affect the latter's oligomerization process, by regulating a conformational change in Orb2 which now makes it an amenable unlocked substrate on which other chaperones or other regulatory proteins might act to drive the oligomerization process.

## Mrj is needed in the mushroom body for the regulation of long-term memory

As our observations imply Orb2 oligomerization is dependent on Mrj, we next asked if Mrj is needed for long-term memory. We took the wild-type and Mrj knockout animals and subjected them to courtship suppression-based memory assay (Fig 6A). Male flies when subjected to repeated rejections by mated females, suppress their courtship behavior post-training. Compared to wild-type animals, mutants defective in their ability to retain memory show a faster recovery of the courtship behavior suggesting a decrease in their capacity to remember. At post-training time points of 2 h and 12 h, we could not detect any significant difference in memory scores between the wild-type and Mrj knockout groups, suggesting a normal formation of early and intermediate-term memory. However, at the 16-h and 24-h time points, the memory score became significantly reduced for the Mrj knockout in comparison to the wild type (Fig 6B). This suggests Mrj is needed for regulating long-term memory in *Drosophila*. We also did a control experiment where we crossed the Mrj knockout flies with another P element inserted line (KG04490) near the Mrj locus which does not disrupt the Mrj locus. The progeny which was heterozygous for Mrj knockout were tested for memory at 16- and 24-h time points in comparison with the wild-type flies. Here, we did not notice any significant difference between the memory scores of the 2 groups (S6A Fig), suggesting loss of 1 copy of Mrj is not sufficient to cause a memory deficit.

We next asked if Mrj is expressed in brain structures that are relevant for memory. Towards this, we used the Mrj knockout line, where we introduced a Gal4 cassette in the locus. On using this line to express CD8GFP, we observed labeling in the mushroom body lobes (Fig 6C). As, in the *Drosophila* brain, the mushroom body is the center for the formation and storage of memory, we enquired if Mrj is needed in the mushroom body to regulate memory.

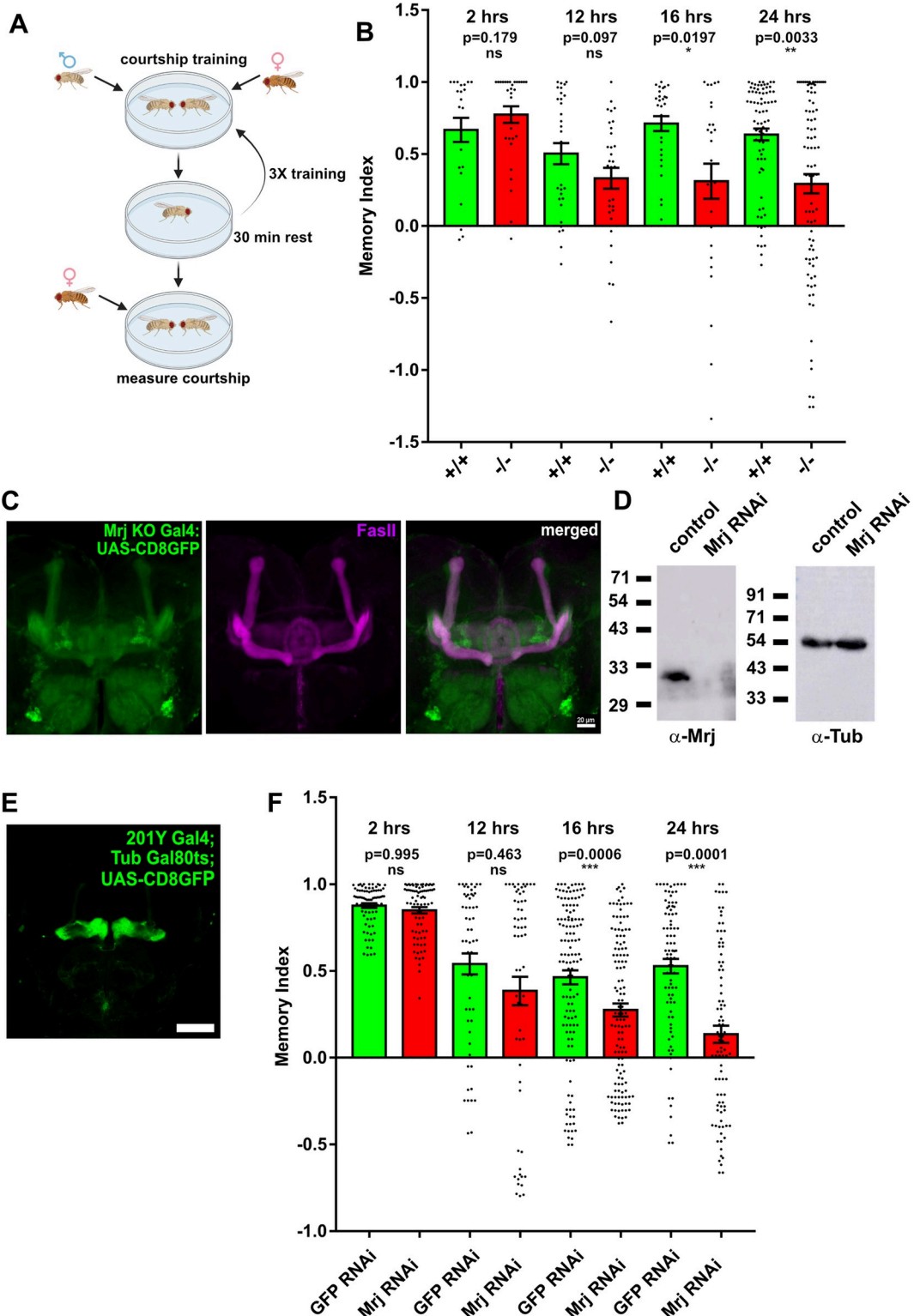

**Fig 6. Mrj regulates long-term memory.** (**A**) Schematic of male courtship suppression-based memory assay. (**B**) Mrj knockout flies show significant memory deficit in comparison to the wild-type flies from 16 h onwards. Data are represented as mean ± SEM and Mann–Whitney U test is done to test for significance. (**C**) Representative image of the expression pattern of Mrj KO Gal4 visualized using the reporter CD8GFP (green) and counterstained with anti-FasII antibody (magenta) shows the expression of GFP in mushroom body neurons. Scale bar is 20 micron. (**D**) Confirmation of knockdown of Mrj by pan-

neuronal expression of an Mrj RNAi line using a western blot with anti-Mrj antibody. Anti-α-Tubulin antibody was used as the loading control. (**E**) Representative image of the expression pattern of 201Y Gal4 visualized using the reporter CD8GFP. (**F**) Knockdown of Mrj in specific mushroom body neurons using 201Y Gal4 causes a significant memory deficit in comparison to the control from 16 h onwards. Data are represented as mean ± SEM and Mann–Whitney U test is done to test for significance. The data underlying this figure are available at: https://figshare.com/s/f5d913a0a289339ee16b.

Toward this, we used an Mrj RNAi line. We first checked for the ability of the Mrj RNAi line to knock down Mrj in the brain. On inducing Mrj RNAi using a pan-neuronal Elav Gal4 driver, we could not detect Mrj in the RNAi fly head extract in comparison to the control (Fig 6D). This indicated the effective knockdown of endogenous Mrj to non-detectable levels. Driving the same Mrj RNAi line using a glial cell-specific driver Repo Gal4, did not show any decrease in Mrj levels (S5H Fig). We also took advantage of the Mrj KO Gal4 line by using it to drive NLS-GFP and immunostained these brains with a glia-specific anti-Repo antibody. Here, we could see cells that were positive for both Repo and Mrj Gal4 driven GFP (S5I Fig), suggesting Mrj apart from neurons is most likely also expressed in glial cells. On our observation of not detecting any Mrj on driving the Mrj RNAi line using Elav Gal4, one explanation can be the majority of the endogenous Mrj is probably contributed from the neurons. Another possibility is, it is reported that Elav Gal4 expresses in neuroblasts and glial cells at earlier developmental stages [76], and so the Elav Gal4 is also knocking down Mrj in these cells along with the neurons causing a drastic decrease of Mrj levels. Using this Mrj RNAi line we now asked, if Mrj is specifically required in the mushroom body neurons for memory. We used a mushroom body-specific 201Y Gal4 line which extensively expresses in the γ lobes along with some neurons of the α/β lobes (Fig 6E) to knock down Mrj in mushroom body neurons. Previously, the expression of Orb2 transgenes using this 201Y Gal4 line was found to be sufficient to rescue memory deficits in Orb2 mutants [18]. Hence, this Gal4 line provides us with the advantage of knocking down Mrj only in the Orb2-relevant neurons. Though earlier we could not detect any gross developmental defects in Mrj knockout flies, we wondered if there might be fine undetected developmental defects. To rule out memory deficiency caused due to such defects, we decided to perform the Mrj knockdown in the adult stage after all development has taken place. Towards this, we coupled the 201Y Gal4 line with a temperature-sensitive Gal80 ts [77,78] and used this to perform the Mrj knockdown with the inducible Mrj RNAi line. These flies were allowed to lay eggs, grow, and eclose at 22°C temperature. At this temperature, the Gal80 would suppress the transcriptional activity of Gal4 and the Mrj RNAi would not be induced. Post eclosion, these flies were shifted to 30°C, where the Gal80 becomes inactive resulting in restoration of the transcriptional activity of Gal4 causing the Mrj RNAi to knock-down Mrj in the mushroom body neurons. We next subjected these mushroom body-specific Mrj knockdown animals to courtship suppression-based memory assay as described earlier (Fig 6A). At post-training time points of 2 h and 12 h, we could not detect any significant difference in memory scores between the control and Mrj knockdown groups, suggesting a normal formation of early and intermediate-term memory. However, at the 16-h time point, the memory score became significantly reduced for the Mrj knockdown animals in comparison to the control animals (Fig 6F). This difference became even greater at the 24-h time point where the memory score for Mrj knockdown decreased drastically compared to the control (Fig 6F). As a control for the same experiment, we left the crosses of the 201YGal4; Tub Gal80 ts with the control and Mrj RNAi lines to grow and eclose at 18°C, where the Gal80 ts will not be inactivated and hence the RNAi will not be induced. These flies when tested in the memory assays showed no significant difference in the memory scores at any of the time points including 16 and 24 h (S6B Fig). Together these experiments suggest Mrj is needed in mushroom body 201Y-specific neurons for regulating long-term memory in *Drosophila*.

## Mrj regulates the association of Orb2A with translating ribosomes

Since long-term memory is dependent on protein synthesis, and the knockdown of Mrj in mushroom body neurons causes a long-term memory deficit, we asked if Mrj plays any role in this protein synthesis. We first addressed if Mrj can associate with ribosomes. We cotransfected Mrj with Rpl18-Flag, which was earlier found to be incorporated in assembled ribosomes and can be used to pulldown polysomes [41]. On immunoprecipitating Rpl18 with an anti-Flag antibody, we could detect Mrj in the immunoprecipitate (Fig 7A). We also performed a reverse immunoprecipitation using an anti-HA antibody to pull down Mrj-HA and in this immunoprecipitate, we could detect Rpl18 using anti-Flag antibody western (S6C Fig), suggesting Mrj most likely interacts with the translating ribosomes.

We next performed polysome fractionation of Mrj expressing S2 cell extract in a 5% to 40% sucrose gradient. On probing the fractions, we found Mrj to be present in the polysome fractions (Fig 7B). However, on doing the fractionation for EDTA treated extract, which disassembles the polysomes, we observed Mrj is still present in all the fractions but now getting more enriched and shifting to the heavier fractions (Fig 7C). One possible explanation for this is, Mrj forms oligomers which are probably of similar density as the polysomes and so we can detect Mrj in these fractions. Since we observed the formation of Mrj oligomers with recombinant Mrj, we loaded the recombinant Mrj on an identical sucrose gradient and fractionated it. On probing the different fractions for the presence of Mrj, we noticed while Mrj is distributed across different fractions, it is still not there in the fractions where the heavier polysomes come (S6D Fig). We also noticed the addition of EDTA does not change the distribution of recombinant Mrj, unlike what we observed with cell extract. One possible reason explaining the shift of Mrj in cell extract to heavier polysome fractions is there is some polyribosome-associated stabilization of Mrj which prevents it from going to the heavier fractions. Taken together the observations of Mrj and Rpl18 interaction, the presence of cellularly expressed Mrj in polysome fractions and the absence of recombinant Mrj in the same fractions, suggest a likelihood of association between Mrj and the polysomes.

We next checked if Mrj played any role in global protein synthesis. For this, we used the Puromycin incorporation assay to compare protein synthesis between wild-type and Mrj knockout brains [79,80]. Puromycin gets incorporated into translating proteins due to its similarity with tRNA and blocks further incorporation of amino acids. The amount of Puromycin incorporation can be visualized using immunostaining with an anti-Puromycin antibody. Using this assay, we did not observe any significant difference in Puromycin incorporation between the brains of wild type and Mrj knockout (Fig 7D and 7E). This suggests the absence of Mrj does not cause any gross defect in general translation.

We next addressed the effect of Mrj on the association of Orb2A with the translating ribosomes. We transfected S2 cells with Orb2A along with and without Mrj and used these cells to perform polysome fractionation. On assaying the different fractions with anti-Orb2 antibody, we observed that in the presence of Mrj, more Orb2A is present in the polysome fractions (Fig 7F and 7G). This indicates that in the presence of Mrj, the association of Orb2A with the translating ribosomes is enhanced, however, if this is a consequence of increased Orb2A oligomers due to Mrj or caused by interaction between polysome associated Orb2A and Mrj will need to be tested in future.

## Discussion

One key mechanism involved in the regulation of the persistence of memory is the prion-like activity of CPEB in Aplysia [16,17,25–27,81] and Orb2 in *Drosophila* [18,40,41,43,82]. In this work, we start with an immunoprecipitation screen and identify interactors belonging to the

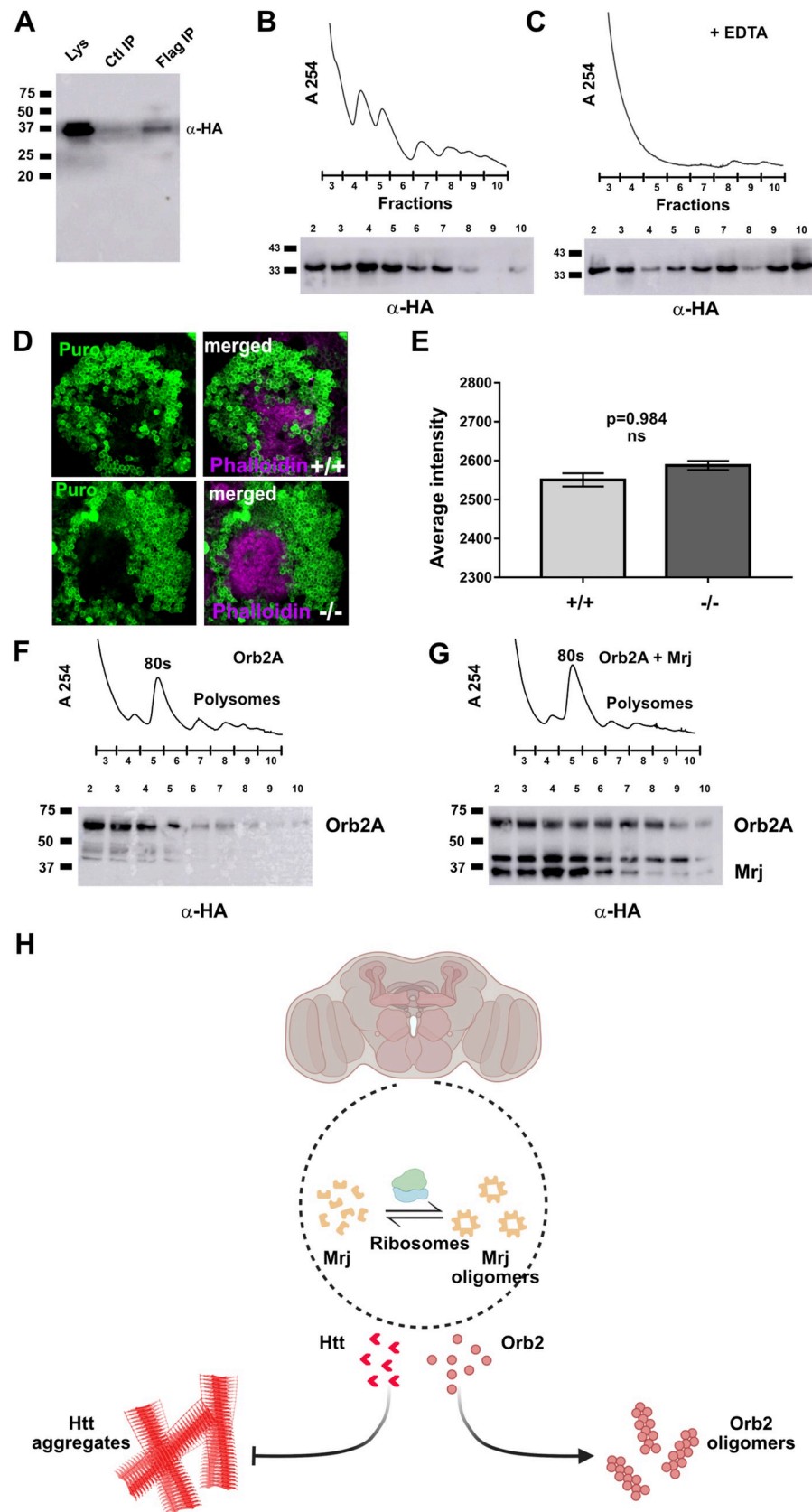

**Fig 7. Mrj interacts with ribosomes and helps in regulating Orb2A's association with polysomes. (A)** Immunoprecipitated Rpl18-Flag which is incorporated in ribosomes pulls down Mrj, suggesting a possible association of Mrj with ribosomes. **(B)** Upper panel shows a representative polysome profile of lysate obtained from cells expressing Mrj-HA. The numbers in X-axis represent the fraction number. The lower panel is a western blot of individual fractions, showing the presence of Mrj in polysome fractions. **(C)** Upper panel shows a representative polysome profile of EDTA-treated lysate obtained from cells expressing Mrj-HA. The numbers in X-axis represent the fraction number. The lower panel is a western blot of individual fractions, showing the shift of Mrj to heavier fractions in comparison to non-EDTA treated lysate. **(D)** Representative images of mushroom body Kenyon cells immunostained with anti-Puromycin antibody to detect newly translated proteins. Magenta shows the counterstain with Phalloidin. **(E)** Quantitation of anti-Puromycin staining intensity shows no significant difference between the wild type (+/+) and Mrj knockout (-/-). Data is represented as mean ± SEM and significance is tested using two-tailed Student's unpaired $t$ test. **(F)** Upper panel shows a representative polysome profile of lysate obtained from cells expressing Orb2A-HA. The numbers in X-axis represent the fraction number. The lower panel shows a western blot detecting Orb2A in individual fractions. **(G)** Upper panel shows a representative polysome profile of lysate obtained from cells co-expressing Orb2A-HA and Mrj-HA. The numbers in X-axis represent the fraction number. The lower panel shows a western blot detecting Orb2A and Mrj in individual fractions. In comparison to only Orb2A, in presence of Mrj, more Orb2A is detected in polysome fractions. **(H)** Schematic of the model originating from this work. We hypothesize association of Mrj with Ribosomes is probably stabilizing it and regulating its oligomerization. Mrj acts as an inhibitor of aggregation of pathogenic Huntingtin (Htt) and positively regulates the formation of Orb2 oligomers. The data underlying this figure are available at: https://figshare.com/s/f5d913a0a289339ee16b.

Hsp40 and Hsp70 families of proteins. On screening these interactors in a heterologous yeast Sup35-based prion conversion assay, we identify Mrj as a protein that is regulating the conversion of Orb2A from non-prion to prion-like oligomeric form. The mammalian homolog of Mrj, DNAJB6 was earlier reported to be acting as an oligomeric chaperone interfering with the aggregation of Huntingtin aggregates [83]. DNAJB6 was also observed to be up-regulated in the astrocytes of Parkinson's patients and help in suppressing α-Synuclein aggregation, suggesting an overall protective mechanism against aberrant aggregation of proteins causing neurodegeneration [84,85]. Like its mammalian homolog, we found *Drosophila* Mrj to be an oligomeric protein interacting with Hsp70 chaperones, which can decrease Huntingtin aggregates in cells. This along with previous reports of *Drosophila* Mrj rescuing effects of polyglutamine aggregates [68,86] suggests that *Drosophila* Mrj acts similarly to its mammalian homolog as a chaperone.

The DNAJB6 knockout mice were reported to be lethal at the mid-gestational stage at embryonic 8.5 days due to failure in chorioallantoic fusion causing defective placental development [49]. In the knockout, the Keratin filaments had undergone collapse due to the formation of keratin inclusion bodies which disrupted the chorionic trophoblasts leading to cellular toxicity [87]. Apart from the keratin filament disorganization these knockouts also exhibited defects in Actin cytoskeleton organization, misexpression of E-cadherin and β catenin, and disorganization of extracellular matrix (ECM) [88], severe neural defects including exencephaly, defect in neural tube closure, reduced neural tube size, and thinner neuroepithelium [89]. All of this suggests that DNAJB6 is an essential gene in mammals. DNAJB6 was previously implicated in LGMD where patients with missense mutations in the gene showed the presence of rimmed vacuoles and inclusion bodies in the muscle [90–92]. In contrast to DNAJB6-associated phenotypes, we found *Drosophila* Mrj knockout to be homozygous viable and having no signs of defects in the development of the brain, cytoskeleton, and muscle. There was no difference with the wild-type control in terms of life span and accumulation of Ref(2)P and Ubiquitinated proteins, which are markers of aging and aggregates in the brain. While the mice knockout for DNAJB6 is embryonic lethal what makes the *Drosophila* Mrj knockout to be so different from the observed phenotype of mammalian Mrj knockout? DNAJB6 was found to be an interactor of keratin protein and the DNAJB6 knockout exhibited keratin aggregates which might have contributed to the lethality. In *Drosophila*, since there is no keratin homolog present, we speculate that this probably helps the Mrj knockout to escape lethality.

One caveat here is loss of Mrj in *Drosophila* may still result in a protein aggregation-related disease phenotype, probably under a sensitized condition of certain stresses which is not tested in this manuscript.

We find Mrj knockout to have reduced Orb2 oligomers in comparison to wild type suggesting a requirement of Mrj in regulating the formation of endogenous Orb2 oligomers. Our findings indicate that Mrj's behavior towards Orb2 is very different from its known protective, anti-aggregating function against pathogenic proteins associated with neurodegenerative diseases. While for both Huntingtin and Synuclein, deletion of Mrj/DnaJB6 caused an increase in their aggregation [83,84,93], in the case of Orb2, deletion of Mrj decreased its aggregation as evidenced from an increased amount of Orb2 in the Triton X-100 soluble state and reduced Orb2 oligomers in both 0.1% SDS and 2% SDS soluble states. This suggests the same chaperone may have different effects on the regulation of aggregation for pathogenic against non-pathogenic functional amyloids/prions (Fig 7H).

How does the Mrj-dependent regulation of the prion-like nature of Orb2, compare with other studies of the effect of Hsp40 chaperones with prions and RNA-binding proteins? Studies in yeast showed the maintenance and propagation of 3 prions associated with Rnq1, Ure2, and Sup35 are dependent on an Hsp40 family protein Sis1 [94–96]. Sis1 acts as a disaggregase in collaboration with Hsp104 and Hsp70 by helping in the fragmentation of the oligomers of these prions to generate propagating seeds [97]. Another Hsp40 family protein Ydj1 together with Hsp70 has been found to inhibit the formation of the Ure2 prion state in yeast by delaying its aggregation to form amyloid fibrils [98,99]. Apart from the regulation of prions, the chaperones Sis1 together with Hsp104 and Hsp70 regulate the dispersal of the heat-induced aggregates of an RNA-binding protein Pab1 [100]. Overexpression of mammalian Mrj/DnaJB6 was able to cure Ure2 prions by helping in the dissolution of the aggregates in cells [101,102]. This suggests the chaperones including Mrj have an anti-aggregation role for prion aggregates whereas, in the case of Orb2, this is promoting its aggregation. One question here is why Mrj behaves differently with Orb2 in comparison to other amyloids. Orb2 differs from other pathogenic amyloids in its extremely transient existence in the toxic intermediate form [39]. For the pathogenic amyloids, since they exist in the toxic intermediate form for longer, Mrj probably gets more time to act and prevent or delay them from forming larger aggregates. For Orb2, Mrj may help to quickly transition it from the toxic intermediate state, thereby helping this state to be transient instead of longer. An alternate possibility is post-transition from the toxic intermediate state, Mrj stabilizes these Orb2 oligomers and prevents them from forming larger aggregates. This can be through Mrj interacting with Orb2 oligomers and blocking its surface thereby preventing more Orb2 from assembling over it. Another difference between the Orb2 oligomeric amyloids and the pathogenic amyloids is in the nature of their amyloid core. For the pathogenic amyloids, this core is hydrophobic and devoid of any water molecules; however for Orb2, the core is hydrophilic. This raises another possibility that if the Orb2 oligomers go beyond a certain critical size, Mrj can destabilize these larger Orb2 aggregates by targeting its hydrophilic core.

The only Hsp40 chaperone which was found similar to Mrj in increasing Orb2's oligomerization is the yeast Jjj2 protein. In Jjj2 knockout yeast strain, Orb2A mainly exists in the non-prion state, whereas on Jjj2 overexpression the non-prion state could be converted to a prion-like state. In S2 cells coexpression of Jjj2 with Orb2A lead to an increase in Orb2 oligomerization [59]. However, Jjj2 differs from Mrj, as when it is expressed in S2 cells, we do not detect it to be present in the polysome fractions (S6E Fig).

Our observations of long-term memory deficit in Mrj knockout and on knocking it down in the specific mushroom body neurons suggest Mrj plays a role in these neurons for the regulation of long-term memory. Some previous studies indicated that chaperones in the context

of neurodegenerative diseases might help in the regulation of memory. A DnaJ domain-containing protein Droj2 has been suggested to be a negative regulator of memory in the *Drosophila* model of Alzheimer's disease [103]. ER chaperone Hspa5 and small heat shock protein 22 (sHsp22) were found to improve the impairment in spatial learning and memory deficit in a mice Tauopathy model [104,105]. Through our work, we implicate the chaperone Mrj to play a crucial role in the regulation of long-term memory in a non-stressed and non-disease condition.

In this study, we also try to find a plausible mechanism of Mrj-dependent memory regulation. As protein synthesis is vital for long-term memory regulation and Orb2 is a translation regulator, we asked if Mrj plays any role in this. We observed that Mrj interacts with ribosomes and is present in the polysome fractions. This observation is also supported by proteomics-based interactome analysis of human DNAJB6, where it was found to interact with several ribosomal proteins [106]. DNAJB6 exists in an equilibrium between its monomers and oligomers and interfering with its oligomerization reduces its anti-aggregation property on Htt aggregates [65,107]. We observe that on disrupting the polysomes, Mrj migrated to heavier fractions. Based on this, we hypothesize the association of Mrj with the polysomes is probably regulating its oligomerization and stabilizing them and preventing it from forming bigger oligomers.

There are also examples from yeast where DnaJ domain-containing Hsp40 family proteins, Zuotin1, Sis1, and Jjj1, and Hsp 70 family proteins Ssa1 and Ssz1 have been reported to associate with ribosomes [108–110]. Zuo1 and Ssz1 together are referred to as ribosome-associated chaperones (RACs), and together with Ssb, they interact with the newly translated protein coming out of the ribosome tunnel and help in its folding. Since the loss of Mrj does not cause any deficit in global protein synthesis, our findings suggest that Mrj is not a "generalist" but a "specialist" who is most probably involved in regulatory roles for only certain substrates. In this case, Mrj regulates the oligomerization of Orb2A and in its presence, we see increased amounts of Orb2A being present in the polysome fractions. This indicates that in the presence of Mrj, the association of Orb2A with the translating ribosomes is enhanced; however, if this is a consequence of increased Orb2A oligomers due to Mrj or caused by interaction between polysome-associated Orb2A and Mrj will need to be tested in the future.

While we have identified Mrj as the chaperone regulator of Orb2 oligomerization, Orb2 is probably not the only substrate of Mrj. Another known RNA-binding protein interactor of Mrj is Hrb98DE, the *Drosophila* ortholog of human hnRNPA1 and hnRNPA2B1 associated with inherited myopathies [111]. There might also be other RNA-binding proteins as the substrate of Mrj. Different DnaJ domain proteins have been reported to interact among themselves and form functional disaggregation complexes [112]. Whether Mrj also interacts with other Hsp40 family proteins and forms functional complexes in the regulation of Orb2 and long-term memory needs to be addressed in the future. In this context, our immunoprecipitation-based screen identified 5 other Hsp40 family proteins and 4 Hsp70's but these proteins could not convert Orb2A PrD from its non-prion to prion-like state in the chimeric Sup35-based assay. What these interacting proteins are doing through their interaction with Orb2A and if they are even expressed in the Orb2 relevant neurons will need to be tested separately and will be the subject of our future studies.

Can our observation in *Drosophila* also be relevant for higher mammals? We tested this with human DnaJB6 and CPEB2. In mice CPEB2 knockout exhibited impaired hippocampus-dependent memory [22], so like *Drosophila* Orb2, its mammalian homolog CPEB2 is also a regulator of long-term memory. In immunoprecipitation assay, we could detect an interaction between human CPEB2 and human DnaJB6 (S6F and S6G Fig), suggesting the feasibility for DnaJB6 to play a similar role to *Drosophila* Mrj in mammals. However, as the human DnaJB6

level was observed to undergo a reduction in transitioning from ES cells to neurons [83], how this can be reconciled with its possible role in the regulation of memory. We speculate, such a reduction if it is happening in the brain will occur in a highly regulatable manner to allow precise control over CPEB2 oligomerization only in specific neurons where it is needed and the reduced levels of DnaJB6 is probably sufficient to aid CPEB oligomerization. Alternatively, there may be additional chaperones, possibly including the human homologs of the other DnaJ interactors identified here (S2 Data), that may function in a stage-specific manner and be able to compensate for the decline in levels of DNAJB6.

## Methods and materials

### Construction of library to express Hsp40 and Hsp70 proteins in S2 cells and yeast

The *Drosophila* Hsp40 and Hsp70 proteins were PCR amplified using gene-specific primers (Table 1) and were cloned into a Topo-D-Entr vector using TopoD-Entr cloning kit (Invitrogen). Sequence-confirmed constructs were then transferred to destination vector pUASg-HA (Hsp40 genes) with 3X HA tag in the C terminal [113] and pUASt-ccdB-FLAG (pTWF) with 3X FLAG tag in the C terminal (For Hsp70's) using LR clonase, and these clones were used for all S2 cell-based experiments. Mrj-RFP construct was made by transferring Topo-D-Entr-Mrj construct to the destination vector pAc5.1-ccdB-RFP (pAWR) using LR clonase. For expression of Hsp40s and Hsp70s in yeast, Topo-D-Entr clones were cloned in destination vector pAG424-GAL-ccdB-HA with 3X HA tag in the C terminal. Orb2APrD-Sup35C chimera was prepared by transferring a Topo-D-Entr construct of Orb2A (corresponding to N terminal 162 amino acids) to the destination vector pAG415-ADH1-ccdB-SUP35C using LR clonase.

### Construction of truncated Orb2A and Orb2B constructs to express in S2 cells

Truncated Orb2A and Orb2B fragments amplified by the designated primers (Table 2) were cloned in Topo-D-Entr and then transferred to the destination vector pUASt-ccdB-GFP (pTWG) using LR clonase.

### Construction of Mrj mutant construct

The Topo-D-Entr Mrj construct was mutated (HPD to QPD) with a site-directed mutagenesis kit from NEB using the following primers (Table 3). This mutated construct was then used in LR cloning with pAWR vector to construct the Mrj31Q-RFP.

### Construction of Baculo virus for expressing Mrj and Orb2

The Mrj-HA fragment was excised from pUASg-Mrj-HA construct with NotI and NheI digestion and cloned in pFastBac1 digested with the same enzymes. For untagged Orb2A, the Orb2A fragment was PCR amplified and cloned in pFastBac1, for the GFP-His tagged Orb2A, the same was cloned in pFastBac-GFP-His vector. These pFastBac-based constructs were next transformed in DH10Bac cells and positive colonies were screened using X-Gal based Blue white screening. While colonies were further grown and Bacmid was purified. These Bacmids were next transfected in Sf9 cells using Cellfectin (Invitrogen) to generate the Baculo virus, which was confirmed for expression using western blots.

**Table 1. List of primers used to amplify Hsp40 and Hsp70 genes.**

| | Gene | Location | Primer set for Topo d Entr cloning | |
|---|---|---|---|---|
| | | | **Forward** | **Reverse** |
| 1. | CG2790 | 60E8 | CACCAACATGCGTTGCTATTACGAGGAGCTCGA | CTTGTTGCGCTTGCCCTTAGCTTTGCCA |
| 2. | CG2887 | 9D1 | CACCAACATGCCTAAAGACTACTACAAGATA | GTTTTGCAGAAGTCTGTCGAGCGAGGA |
| 3. | CG2911 | 83B6 | CACCAACATGCCCGACTTCTATGAACTCTT | TTATTTACAACTTGCGGCTTGTTTCA |
| 4. | CG4164 | 21C2 | CACCAACATGCAGCTTATCAAGTGCTTGGT | CAGTCCATTGTATATGCGATTGATGGA |
| 5. | CG3061 | 88A4 | CACCAACATGGACGGAAACAAGGACGA | TGTGATTAAATACTTTTGCAGATTCTCGCA |
| 6. | CG5001 | 21F2 | CACCAACATGGGTAAGGACTACTACAAAAT | TAACATGTCCTTGAGCACCTCCTTCT |
| 7. | CG6693 | 86D8 | CACCAACATGTCGACGCTGGAGCTGTGCGA | GTTTTTGCCCTTCTCCACACGGCCCGCCT |
| 8. | CG7130 | 79B2 | CACCAACATGGGTAAGGATTACTACAAGAT | GTCCGAGGAGGATCCATCGCCGT |
| 9. | CG7133 | 79B2 | CACCAACATGAGCGATGTCTACGAAGATCACT | ATTTTTCAAGCTTCCTTTAAAAGATTGT |
| 10. | CG7387 | 66B11 | CACCAACAATGTTTCGTTTAAAATTATTTCGCAGT | GTGACTCAAATTGCCCGCCTCCGT |
| 11. | CG7556 | 18A5 | CACCAACATGATGATGCGACCGGAGCTGACGCT | ACTGATCTCGTACTGATATGCATCGTCCGA |
| 12. | CG7394 | 68C13 | CACCAACATGGCGAGCTCCGTAATTCT | CTTCGCTTTGTCCAGAAAGTCTTTGGCCT |
| 13. | CG7872 | 13D4 | CACCAACATGCGGACGCACGGACTACGGTTGGT | ATCCTCAAAGGTGATGCGGCCGGGT |
| 14. | CG8476 | 87E2 | CACCAACATGTTGAGGACCATCCAGACCGGA | TGCAGATGGATGGATTGGATGAGTTAT |
| 15. | CG8531 | 50E6 | CACCAACATGGCATCGGATCACGATGAGTCCGAT | TCCTGACAGCGGTTGTCGCAGCGGCA |
| 16. | CG9828 | 34A7 | CACCAACATGGACAACCTAAATTTATACGA | AGCCGTCTGGCACTGTACGCCCTCAAAGT |
| 17. | CG10375 | 95B1 | CACCAACATGTCCTCCTCGTCGGCCAGAGACCA | TCGCGACTCGGGCTTAAATTTTGGTGGA |
| 18. | CG10565 | 78B2 | CACCAACATGACGAGCGGTACGGTAGCAAC | TTTGACCGCCGCCTGTGCCTCCTTCTTGGA |
| 19. | CG11035 | 84E9 | CACCAACATGTACCAGCACAGCCCAATGTTGT | AGATTGCTTCCCGACTAGTTTTTGTTCT |
| 20. | CG12020 | 62B4 | CACCAACATGAACCGGCCGGAGCTGGACTACTA | AACCATGTTACGTTCCTCCTCCTCCT |
| 21. | CG14650 | 82B1 | CACCAACATGGCGACTCGTCAGGACGACGA | GTTACGTCGCGTTCGACGACGAGCATTTCC |
| 22. | CG17187 | 86D8 | CACCAACATGGCTAGCAAAAAGTACAGTGACGT | TTCGCCCTCCTCGTCCTTCATCATCTGCT |
| 23. | CG30156 | 42E1 | CACCAACATGGGAATCCTGCAGGCGAGGA | GACTGGCACGTCCTGATTCAAAGACTCGT |
| 24. | CG32641 | 11E3 | CACCAACATGGAGGAGGACTACTATATGATCCT | ATGATTATTATTCGATCCGCTTAATGCCT |
| 25. | CG43322 | 27D4 | CACCAACATGTGGTTGGCTGGGAATCTGTTCA | CAATAGTTTGCTAATTAGGGAAAATAAACT |
| 26. | CSP | 79D3 | CACCAACATGAGCGCACCTGGCATGGACA | TATACCTGGCGTGTAAGTGGTCTGCTCCGT |
| 27. | DnaJ1 | 64E5 | CACCAACATGGGCAAAGACTTCTACAAGATT | GTTGGGCAGCAGCTCGGACAGCTGATTCT |
| 28. | DnaJ60 | 60C1 | CACCAACATGCTAAGGTTGTGCCTTCCAACT | CTTAGCCGAAGACTGGTTGAAAGGACT |
| 29. | DroJ2 | 87E8 | CACCAACATGGTTAAGGAGACTGGATATTATGA | ACTCGATGTGCACTGCTGGACGCGTGGGCCA |
| 30. | Hsc20 | 72D10 | CACCAACATGAGTAGAGTTATTAATGGCTTTA | GCTGCCCAGCAAACTTTGTTGCTTCTGT |
| 31. | Jdp | 99F7 | CACCAACATGAGCGCCGTTGATGCTATA | AAAAACAAGGCCAAGCGTTGACGCAGT |
| 32. | Mrj | 52F8 | CACCAACATGGTTGACTACTATAAAATTTT | TTGAAGGGAGCCCATCACAGTCTTTGATTTCA |
| 33. | P58IPK | 85D27 | CACCAACATGGCCTTGCCATTGAGTGACCTGCT | ATTGAAGTGGAACTTGAACTGGAACGGCGA |
| 34. | Sec63 | 65F7 | CACCAACAATGGCGGGTCAAAAGTTTCAGTATGA | AGATACGTCCGATGAGGAGGACGGCGAA |
| 35. | Tpr2 | 36A2 | CACCAACATGTTGTACATTGCCGAGGAGA | GAACTCAAAGTTGAACGATGAATTATT |
| 36. | Wus | 15C4 | CACCAACATGGGCACCACGCAGACAGTATCT | AAGGACGATGGCCTCCTCCTCCTGGTA |
| 37. | L(2)tid | 59F6 | CACCAACATGATGATTTCGTGTAAAAAATT | GTTGAACATGGACTTGATCTTGCTTATGAA |
| 38. | Luciferase | - | CACCATGGTCTTCACACTCGAAG | TTACGCCAGAATGCGTTCGCACA |
| 39. | Hsc70Cb | 70C15 | CACCAACATGTCCGTGATTGGCATCGAT | CTCCACTTCCATGGAGGGATCGTTTC |
| 40. | Hsc70-4 | 88E4 | CACCAACATGTCTAAAGCTCCTGCTGTTGGT | GTCGACCTCCTCGATGGTGGGGCCA |
| 41. | Hsp70Aa | 87A2 | CACCAACATGCCTGCTATTGGAATCGATCT | GTCGACCTCCTCGACCGTGGGTCCAGA |
| 42. | Hsc70-1 | 70C9 | CACCAACATGAAGCATTGGCCCTTCGAGGT | ATCAACCTCCTCAATGGTTGGACCCGA |

## RT-PCR of Mrj isoforms

Total mRNA was isolated from fly heads of the desired genotype and cDNA was synthesized from mRNA isolated from fly heads by clontech TAKARA cDNA synthesis kit using the

**Table 2. Primers to amplify truncated Orb2A and Orb2B constructs.**

|   | Fragment | Forward | Reverse |
|---|----------|---------|---------|
| 1 | Orb2AΔ162 | CACCATGGGTGGCCTGCCGAAT | ACACCAGCGAAAGGGGACCGCACG |
| 2 | OrbA325 | CACCATGTACAACAAATTTGTTA | TGGCCAATCGACGACCAATGGCCCGA |
| 3 | Orb2B478 | CACCATGGACTCGCTCAAGTTAC | TGGCCAATCGACGACCAATGGCCCGA |

manufacturer's protocol. This cDNA was further used as the template to carry out PCR using NEB Phusion polymerase to screen the Mrj isoforms present in the fly brain using the iso-form-specific primers (Table 4).

## S2 cell culture and transfection

S2 cells were grown in Schneider's media supplemented with 10% FBS. Transfections of plas-mids were done with Effectene (Qiagen) using the manufacturer's protocol.

## Immunoprecipitation from S2 cells

Transfected S2 cells were lysed in lysis buffer (150 mM NaCl, 50 mM Tris, 1% NP40, and pro-tease inhibitors) were centrifuged at 10,000g for 10 min at 4°C. The supernatant was separated and incubated with antibodies for 2 h at 4°C. Immunoprecipitation was performed by binding this lysate with pre-blocked (with 1% BSA) Protein-A beads for an additional 2 h and further collecting the beads by spinning at 1,000g for 2 min. The beads were next washed 3 times for 10 min with cold lysis buffer. Proteins bound to the beads were eluted by the addition of 0.2 M Glycine (pH 2) for 1 min and further neutralization with 1.5 M Tris (pH 9). The eluates were next mixed with loading dye and processes for western blots. Validation of the Orb2 antibodies is shown in S7A, S7B and S7C Fig. Validation of the Orb2 antibody in the immunoprecipita-tion experiments is shown in S7E and S7F Fig.

## Western blot

The immunoprecipitated samples were boiled for 10 min and were directly loaded and ran on polyacrylamide gels (10%, 12%, or 15% gels according to the molecular weight of the protein of interest) and further electro-blotted onto a PVDF membrane for 3 h at 80 V in cold wet transfer buffer. The membranes were blocked with 5% nonfat dry milk in TBST buffer and incubated with respective primary and HRP-conjugated secondary antibodies. The membrane was washed with TBST and incubated with chemiluminescence reagents. The membrane was next put between 2 transparency sheets and the chemiluminescent signals on it were imaged using a GE Healthcare Lifesciences AI600 Imager.

## Yeast prion assay

A yeast strain bearing a Sup35 knockout rescued by a Sup35 expressing plasmid sup35::HygB pAG426GPD-SUP35 (Mata, leu2-3,112;his3-11,-15;trp1-1;ura3-1;ade1- 14;can1-100;[RNQ+]; sup35::HygB;pAG426GPD-Sup35) was transformed with pAG415 ADH-Orb2APrD-Sup35C/ Leu plasmid and selected on SD Leu⁻ plates. The colonies were grown in YPD media with

**Table 3. Primers for site directed mutagenesis of Mrj.**

| Fragment | Forward | Reverse |
|----------|---------|---------|
| HPDmut Mrj31Q | ACTAAAATGGcagCCAGACAAGA | GCCAGTTTTCGATATGCC |

**Table 4. Primers used for PA, PB, PC, PE, PG, and PH isoforms.**

| Gene | Primer name | Sequence |
|---|---|---|
| Mrj-PA, PB, PC, PD, PE, PG, PH | Mrj fwd | CACCAACATGGTTGACTACTATAAAATTTT |
| | Mrj rev | TTGAAGGGAGCCCATCACAGTCTTTGATTTCA |
| Mrj-PF | Mrj fwd_short | CACCAACATGTACTCCCATTCTTAACAG |
| | Mrj rev | TTGAAGGGAGCCCATCACAGTCTTTGATTTCA |

repetitive media changes/transfer and then plated/streaked on 5-FOA plates, as Ura marker-containing cells would not grow on this selection media resulting in the selection of colonies without the pAG426GDP-Sup35 rescue plasmid. Colonies growing in 5- FOA plates were further re-streaked in Leu⁻ media to check and confirm the presence of pAG415 ADH-Orb2APrD-Sup35C/Leu plasmid. These colonies were also checked by streaking them in YPDG media to confirm they were not petites. Positive colonies were further streaked and grown on YPD media and checked for colonies of red and white color. Isolated red and white colonies were restreaked in SD Ade⁻ media to confirm that only the white colonies but not the red colonies are capable of growing in this. The red colonies which are prion negative were further grown and transformed with pAG424Gal-Hsp40/Hsp70/Trp constructs. As a control, empty 424-Gal vector and 424-Gal-luciferase vectors were used. Colonies were selected on SD Trp⁻ plates. Single colonies were grown in SD Trp⁻ media overnight. The next day cultures were spun down, given 2 washes with PBS, and resuspended in Raf Trp⁻ media by adjusting OD600 to 0.2. Cultures were grown to the mid-log phase and induced by the addition of 2% Galactose. After 24 h of induction, cultures were serially diluted and spotted on ¼ YPD and SD Ade⁻ plates. When the Orb2PrD-Sup35C protein is in non-prion form, it helps the ribosomes encountering the premature stop codon in Ade1-4 to fall off leading to no read-through translation. The colonies for this non-prion state are red and cannot grow in adenine-deficient media. If the Orb2APrD-Sup35C protein gets converted to a prion-like state, it will fail to dislodge the ribosomes encountering the premature stop codons, and as a result, read-through translation will happen and now the cells will be white, which can now grow in adenine-deficient media.

## Generation of recombinant Mrj

Primers MrjFwd_NdeI (GCGGCAGCCATATGTATGGTTGACTACTATAAAAT) and MrjRev_XhoI (GTGGTGGTGCTCGAGTTGAAGGGAGCCCATCA) were designed with a 15 bp overhang from pET28a vector. Mrj was PCR amplified using Q5 polymerase (NEB) using TopoD-Entr-Mrj clone as a template. The amplicon was ligated with a pET28a vector digested with Nde1-HF and Xho1-HF enzymes (NEB) using the Infusion cloning kit (Takara). Ligations were transformed in DH5α cells plated on LB+ Kanamycin plates. Sequence confirmed plasmid was transformed in BL21-DE3 cells and grown in LB+Kan media at 37°C under shaking conditions till an OD of 0.6. The culture was next shifted to 30°C and induced with IPTG (final concentration of 0.5 mM) for 6 h. The cells were next pelleted with centrifugation and lysed in denaturing lysis buffer (8 M Urea, 100 mM $Na_2PO_4$, 10 mM Tris-Cl, 10 mM Imidazole) with sonication. The lysate was centrifuged at 10,000g for 10 min and the supernatant was allowed to bind with 1 ml of Ni-NTA beads (Qiagen) for 1 h. Post binding the beads were washed with denaturing wash buffer (8 M Urea, 100 mM $Na_2PO_4$, 10 mM Tris-Cl, 40 mM Imidazole). The wash buffer was changed from denaturing wash buffer to native wash buffer (100 mM $Na_2PO_4$, 300 mM NaCl, 40 mM Imidazole) by a gradual decrease of Urea in the wash buffer. Mrj protein bound to these beads was next eluted using native elution buffer

(50 mM $Na_2PO_4$, 300 mM NaCl, 300 mM Imidazole). This purified protein was used for raising antibodies and downstream DLS experiments.

## Dynamic light scattering

Recombinant Mrj-His protein of concentration around 10 mg/ml was passed through a 0.45 micron PVDF filter to remove pre-formed aggregates. The purified protein was diluted 1:10 in the native lysis buffer which was earlier used during purification. DLS experiments for consecutive 3 days were performed using a Zetasizer NANO (Malvern Panalytical, Malvern, United Kingdom).

## Recombinant protein-based pulldown assay to check Mrj and Orb2A interaction

A pGEX6P1-Mrj construct was transformed in BL21 (DE3) Rosetta cells. Transformed colonies were grown in LB to an OD of 0.8, induced with 0.1 mM IPTG for 22 h at 12˚C. Post induction, cells were harvested by centrifugation and lysed in lysis buffer (50 mM Tris-HCl (pH 7.4), 500 mM NaCl, 1 mM $MgCl_2$, 0.2% TritonX-100, 10% Glycerol) followed by sonication. Lysate was cleared by centrifugation and was allowed to bind with Glutathione beads for 3 to 4 h. The beads were next washed with wash buffer (50 mM Tris-HCl, 500 mM NaCl, 1 mm DTT, 0.2% TritonX-100, 10% Glycerol). These GST-Mrj bound beads were next incubated overnight with Orb2A-His purified from *E. coli* or Orb2A-GFP-His purified from Sf9 cells. The next day, the beads were again washed and then bound proteins were eluted with elution buffer (50 mM HEPES KOH (pH 7.4), 100 mM KCl, 4 Mm $MgCl_2$, 1 mM CaCl2, 0.1% TritonX-100, 10% Glycerol, 40 mM reduced Glutathione). These eluates were next run on SDS-PAGE and probed with anti-Orb2 and anti-Mrj antibodies.

## Mrj antibody generation

To generate antibodies, purified Mrj was mixed with Freund's complete and incomplete Adjuvants and injected in guinea pig over a 72-day immunization protocol followed by the collection of blood and separation of serum. The serum was used in western blots with wild-type and Mrj knockout fly head extracts to confirm the specificity of the Mrj antibody.

## Generation of Mrj knockout

Upstream gRNA sequence CTTCACTTCACTATCGGTAG[CGG] and downstream gRNA sequence ACTTCGACGGTGTTTGTGAA[TGG] were cloned into the U6 promoter plasmid. Cassette Gal4-RFP containing Gal4, SV40 polyA terminator, 2 loxP sites, 3xP3-RFP, and 2 homology arms were cloned into pUC57-Kan as donor template for repairing Mrj. The targeting gRNA construct along with hs-Cas9 and the donor plasmid were together microinjected into embryos of control strain w[1118]. F1 flies carrying the selection marker of 3xP3-RFP were further validated by genomic DNA PCR and sequencing. This resulting line is a deletion allele of Mrj, with a knockin of the Gal4 cassette in the Mrj locus.

## Soluble-insoluble fractionation

Fly heads of Mrj-KO and wild-type flies were collected and homogenized in lysis buffer (50 mM Tris (pH 7.5), 150 mM NaCl, 0.1% Triton-X-100). These lysates were centrifuged at 14,000g for 15 min at 4˚C and the supernatant was separated as the soluble fraction. The left-over pellet was resuspended in SDS containing lysis buffer (50 mM Tris (pH 7.5), 150 mM NaCl, and 0.1% SDS), sonicated, and centrifuged at 14,000g for 15 min at 4˚C. The

supernatant was separated and designated as the insoluble (0.1% SDS) fraction. The leftover pellet was further resuspended in 2% SDS containing lysis buffer (50 mM Tris (pH 7.5), 150 mM NaCl, and 2% SDS), sonicated, and centrifuged at 14,000g for 15 min at 4°C. The supernatant was separated and designated as the insoluble (2% SDS) fraction.

All the soluble and insoluble fractions were further loaded on an SDS-PAGE or SDD-AGE and were either stained by silver staining or transferred onto a nitrocellulose membrane and probed with an anti-Orb2 antibody.

## Silver staining

Soluble and insoluble fractions were run on a 10% SDS-PAGE. The gel was then washed with distilled $H_2O$ for 5 min followed by putting it in a fixing solution (50% methanol, 12% acetic acid, 0.05% formaldehyde) for 1 h. The gel was then washed 3 times with 50% ethanol for 20 min followed by sensitization with 0.8 mM sodium thiosulfate for 2 min, washing with distilled $H_2O$, and putting in staining solution (20 mg/ml $AgNO_3$, 0.076% formaldehyde) for 15 min in dark. The stained gel was then washed briefly with distilled $H_2O$ and put in a developing solution (6% w/v $Na_2CO_3$, 2% of 0.8 mM sodium thiosulfate, 0.01% formaldehyde) till bands are seen. Then, gel was given a wash with distilled $H_2O$ and put in the stop solution (50% methanol, 12% acetic acid). The gel was then scanned to record the data. Densitometric intensities of individual wells were analyzed with ImageJ software and graphs were plotted using GraphPad Prism software.

## Immunoprecipitation from flies

For immunoprecipitation from fly heads, 100 to 150 adult fly heads were homogenized in lysis buffer containing 150 mM NaCl, 50 mM Tris, 1% NP40, and protease inhibitors. Lysates clarified by centrifugation at 10,000g for 15 min at 4°C were incubated with 5 μl of serum containing antibodies for 2 h at 4°C followed by the addition of pre-blocked (with 1% BSA) Protein-A beads (Repligen) for an additional 2 h. The beads were next washed 3 times for 10 min with cold lysis buffer and bound proteins were eluted by the addition of 0.2 M Glycine (pH 2) for 1 min and further neutralization with 1.5 M Tris (pH 9). These samples were further subjected to SDD-AGE and western blot.

## SDD-AGE protocol

Samples were mixed with 4× SDD-AGE loading dye and were run at 50 V in a 1.3% agarose gel in 1XTAE buffer containing 0.1% SDS. The gel was transferred overnight to a nitrocellulose membrane using capillary transfer with 1× TBS buffer. The membrane was further processed for western blot using an anti-Orb2 antibody.

## Immunostaining and imaging

Transfected S2 cells were plated on coverslip bottom dishes and GFP and RFP coexpressing cells were directly imaged on a Nikon A1R confocal microscope. For immunostaining experiments, these cells were fixed with 4% PFA for 5 min, washed 3× with PBST buffer, followed by blocking for 1 h in blocking buffer (5% normal goat serum (NGS) in PBST buffer). Primary antibodies were prepared in the blocking solution which was added to the cells and incubated for 2 h at room temperature. Post removal of the antibody solution, cells were washed again 3× with PBST buffer and incubated with the secondary antibody tagged with fluorophores in block solution for 1 h at room temperature. Next, these cells were washed 3× with PBST and then imaged on a confocal microscope. For immunostaining of the *Drosophila* brains, a similar protocol was followed, except, the fixation was done for 2 h and the primary and secondary antibody incubations were

**Table 5. List of fly lines.**

| Fly lines | Source |
|---|---|
| 201y GAL4 | BDSC 4440 |
| w[*]; P{w[+mC] = tubP-GAL80[ts]}2/TM2 | BDSC 7017 |
| P{10XUAS-IVS- mCD8::GFP}su(Hw)attP5 | BDSC 32188 |
| UAS GFP RNAI (y[1] sc[*] v[1] sev[21]; P{y[+t7.7] v[+t1.8] = VALIUM20-EGFP.shRNA.4} attP40) | BDSC 41552 |
| UAS MRJ RNAi (y[1] sc[*] v[1] sev[21]; P{y[+t7.7] v[+t1.8] = TRiP.HMS05387}attP40) | BDSC 66921 |
| W1118 (+/+) | NCCS stock collection |
| MrjKO-Gal4 (-/-) | This paper |
| Repo Gal4 | BDSC 7415 |
| UAS-NLS GFP | NCCS stock collection |
| KG04490 | BDSC 13602 |

done overnight, and the brains were mounted with Vectashield between glass coverslip and slide. For the muscle staining, third instar larvae were dissected, fixed for 1 h, washed 3× with PBST, and then stained overnight with fluorophore-tagged Phalloidin.

## Fly stocks, handling, and maintenance

Flies were raised on standard cornmeal food in a 12:12 hours light, dark diurnal cycle at 22°C. The fly lines used in this work are listed in Table 5.

## Adult fly locomotion assay

Single flies were put inside 8-cm long, 0.5 cm diameter transparent tubes using an aspirator, and both sides were sealed with cotton plugs. The tubes were tapped against the table to get the fly to one side, followed by putting the tubes horizontally, next to a scale, and recording a video of the fly's movement. The video was analyzed to measure the speed by measuring the distance traveled to the time.

## Male courtship suppression-based memory assay

Male courtship suppression assays were done as described earlier [40]. Briefly, virgin male flies were made to undergo 3× training with mated females. Each training was of 2 h duration and there was a gap of 30 min where the trainer female was removed. Virgin males which did not undergo any training were used as naïve males for control. Post-training, the male flies were again transferred to new tubes and after different intervals of time, were put into a courtship chamber containing a mated female. Videos of these encounters were recorded for 10 min and further manually analyzed to calculate the courtship indexes per the following formula:

$$Courtship\ index\ (CI) = \frac{Time\ spent\ in\ courtship\ activity}{Total\ time\ duration\ of\ video}$$

The memory indexes for individual flies were further calculated using:

$$memory\ index = \frac{average\ CI\ of\ untrained\ flies - CI\ of\ trained\ flies}{average\ CI\ of\ untrained\ flies}$$

The average memory index was calculated for each genotype at different time points and further plotted using GraphPad prism7.

## Polysome analysis

Polysome analysis was performed on S2 cell extract as per previously published methodology [73]. Briefly, cycloheximide-induced translationally stalled cells were lysed in polysome lysis buffer (300 mM NaCl, 50 mM Tris-Cl (pH 8), 10 mM MgCl$_2$, 1 mM EGTA, 1% Triton-X100, 0.02% Sodium Deoxycholate). The lysate was centrifuged at 10,000g for 10 min at 4˚C, and the resulting supernatant was loaded on a 5% to 45% sucrose density gradient made in resolving buffer (140 mM NaCl, 25 mM Tris-Cl (pH 8), 10 mM MgCl$_2$). This was further spun at 4˚C at 28,000g speed for 2 h on a swing bucket SW41 rotor. The gradient was next fractionated with monitoring of Absorbance at 254 nm using a Biocomp fractionator station. Proteins from all the fractions were precipitated by overnight incubation with 10% trichloroacetic acid at −20˚C. Precipitated proteins were spun down at 12,500g for 15 min and then washed with chilled acetone. Washed protein pellets were resuspended in 2× loading buffer and subjected to western blot.

## Antibodies and other reagents

The following commercially available antibodies were used in this paper: Anti-HA ab9110 (Abcam), Anti-HA ab18181 (Abcam), Anti-Flag F1804 (Sigma), Anti-Huntingtin Mab2166 (Chemicon, Sigma), Anti-GFP 50430-2-AP (Proteintech), Anti-RFP R10367 (Life Technologies), Anti α-Tubulin 66031–1 (Proteintech), Anti-FAS II-1D4 (DSHB), Anti-Repo 8D12 (DSHB), Anti-Puromycin-3RH11(Kerafast), Anti-Ref2P-ab178440 (Abcam), Anti-HA epitope tag clone 16B12 (Biolegend), Multi Ubiquitin chain monoclonal-cloneFK2 14220 (Cayman), Alexa flour 488 Phalloidin-A12379 (Life Technologies), Alexa flour 555 phalloidin-A34055 (Life Technologies), Alexa Fluor 555 anti-mouse A21424 (Life Technologies), Alexa Fluor 488 anti-mouse A11029 (Life Technologies), Alexa Fluor 555 anti-Rabbit A11034 (Life Technologies), Alexa Fluor 488 anti-Rabbit A21429 (Life Technologies).

Anti-Orb2 and anti-Mrj antibodies were raised against recombinant 6X Histidine-tagged full-length protein, in rabbit and guinea pig for Orb2 and guinea pig for Mrj. For immunoprecipitation, protein-A agarose beads (Repligen 102500–03) and RFP trap magnetic beads (Chromo Tech rtma) were used.

## Supporting information

**S1 Fig. Rooted phylogenetic tree with branch length for all the Hsp40 family of proteins along with their predicted domain structures.** This list does not show Auxillin and Rme8, which were not used in the screen here.
(TIFF)

**S2 Fig. Rooted phylogenetic tree with branch length for all the Hsp70 family of proteins along with their predicted domain structures.** Of these, Hsp70Aa, Hsc70-1, Hsc70Cb, and Hsc70-4 were used in the screen.
(TIFF)

**S3 Fig. Representative western blots of 31 Hsp40 proteins, which do not show interaction with Orb2A in the immunoprecipitation screen.** The data underlying this figure are available at: https://figshare.com/s/f5d913a0a289339ee16b.
(TIFF)

**S4 Fig.** (**A**) ClustalO alignment of *Drosophila* Mrj with Human, Frog, Zebrafish, Mouse, and Rat Mrj/DnaJB6. (**B**) RT-PCR using isoform-specific primer sets shows amplification of only the 777 nucleotides long form which corresponds to 259 amino acid isoform of Mrj. (**C**) Representative images of S2 cells expressing Mrj-HA (upper panels) and Mrj-RFP (lower panels). Both constructs show the presence of Mrj in both the nucleus and cytoplasm. Scale bar is of 5 microns. (**D**) DLS experiments with recombinant Mrj over 3 days show its shift to higher sizes with time. (**E**) Left panel shows a representative SDD-AGE from S2 cell lysate coexpressing HttQ138-RFP along with CG7133 and Mrj showed a decreased amount of Htt oligomers in presence of Mrj. The right panel is of a western blot of lysates in SDS-PAGE from S2 cells coexpressing HttQ138-RFP with Mrj and CG7133 showing similar amounts of Htt. (**F**) Representative images of HttQ138-RFP cells coexpressing with CG7133-HA and Mrj-HA suggests a decrease in the Htt aggregates in presence of Mrj. Scale bars are of 5 microns. (**G**) Quantitation of the percentage of HttQ138-RFP expressing cells with aggregates in presence of CG7133 and Mrj suggests a significant decrease of Htt aggregates in presence of Mrj. Data is represented as mean ± SEM and significance is checked using two-tailed Student's paired *t* test. The data underlying this figure are available at: https://figshare.com/s/f5d913a0a289339ee16b. (TIFF)

**S5 Fig.** (**A**) Representative images of S2 cells expressing Orb2A, Orb2A325, Orb2AΔ162, Orb2B, and Orb2B478 constructs tagged with GFP. (**B**) Intensity profile plots from lines drawn over punctae in the cells show colocalization between Orb2A-GFP, Orb2A325-GFP, Orb2AΔ162-GFP, Orb2B-GFP, and Orb2B478-GFP with Mrj-RFP. The Orb2AΔ162-GFP construct does not show colocalization with Mrj-RFP. (**C**) Colocalization quantitation using Pearson's coefficient shows significant colocalization of Mrj-RFP with Orb2AGFP (*n* = 34 ROI's from 4 cells), Orb2B-GFP (*n* = 36 ROI's from 6 cells), Orb2A325-GFP (*n* = 14 ROI's from 5 cells), Orb2B478-GFP (*n* = 32 ROI's from 8 cells) but not with Orb2AΔ162-GFP (*n* = 35 ROI's from 5 cells). Data is represented as mean ± SEM. (**D**) *Drosophila* adult brain image of Mrj knockout (KO) Gal4 driving UAS-CD8GFP showing enriched expression in optic lobes, olfactory lobes, and the mushroom body region. Scale bar is of 50 microns. (**E**) Image from the larval muscle of the same flies show expression of CD8-GFP in the muscle. Scale bar is 20 microns. (**F**) Image of larval muscles from Mrj KO Gal4 driving UAS-NLS-GFP shows the expression in multiple nuclei confirming the expression in muscle fibers. The right panel depicts the merge of NLS-GFP with the DIC image. Scale bar is 20 micron. (**G**) Image of the neuromuscular junction from larvae of Mrj KO Gal4 driving UAS-CD8GFP shows expression in the neuromuscular junction synaptic boutons. Scale bar is 20 micron. (**H**) Driving the Mrj RNAi line in Glial cells using Repo Gal4 shows no difference in Mrj expression in comparison to the control animals as seen in the western blot with anti-Mrj antibody. Anti-α-Tubulin antibody was used as the loading control. (**I**) Representative images from olfactory and optic lobe regions of Mrj KO Gal4 driving NLS-GFP brains immunostained with glia-specific anti-Repo antibody. Arrows in the GFP panel depict cells with only GFP and no Repo expression. Arrows in the Repo panel depict cells with both GFP and Repo expression suggesting the likely expression of Mrj in glial cells. Scale bar is 20 microns. The data underlying this figure are available at: https://figshare.com/s/f5d913a0a289339ee16b. (TIFF)

**S6 Fig.** (**A**) Mrj-/KG04490 flies do not show any significant memory deficit in comparison to the wild-type flies at 16 and 24 h onwards. Data are represented as mean ± SEM and Mann–Whitney U test is done to test for significance. (**B**) Preventing the knockdown of Mrj in specific mushroom body neurons using 201Y Gal4 with Tub Gal80 ts at 18°C does not cause any significant memory deficit in comparison to the control from 16 h onwards. Data are

represented as mean ± SEM and Mann–Whitney U test is done to test for significance. (**C**) Immunoprecipitated Mrj-HA pulls down Rpl18-Flag which is incorporated in ribosomes, suggesting Mrj is associated with ribosomes. The right panel shows the same blot probed with anti-HA antibody and confirming the presence of Mrj-HA in the immunoprecipitate. (**D**) Loading recombinant Mrj on an identical polysome gradient followed by centrifugation and fractionation and probing these fractions with anti-Mrj antibody shows their absence from the heavier polysome fractions. Also, on EDTA treatment the recombinant Mrj does not move to the heavier fractions, unlike our observations with cellularly expressed Mrj. (**E**) Polysome fractionation of Jjj2-HA expressing S2 cells in the absence and presence of EDTA followed by detection of Jjj2 using western blots show its absence in the heavier polysome fractions. (**F**) Schematic of immunoprecipitation assay to check the possibility of interaction between hCPEB2 and hDnaJB6 in HEK cells. The cells were lysed and immunoprecipitation was done with anti-GFP antibody (GFP-Trap beads). The immunoprecipitate was next probed with anti-Flag antibody. (**G**) Representative image of the blot shows a pulldown of DnaJB6 by hCPEB2. The data underlying this figure are available at: https://figshare.com/s/f5d913a0a289339ee16b.
(TIFF)

**S7 Fig.** (**A**) Validation of anti Orb2 antibody raised in guinea pig in western blot with S2 cell lysate from cells expressing Orb2A and Orb2A-GFP. (**B**) Guinea pig anti-Orb2 antibody detects endogenous Orb2B from fly head extract. (**C**) Guinea pig anti-Orb2 antibody detects Orb2A-GFP oligomers from Sf9 cell lysate. (**D**) Validation of anti Orb2 antibody raised in rabbit in western blot with S2 cell lysate from cells expressing Orb2A. (**E**) Representative western blot of immunoprecipitation performed from cells expressing Orb2A with Mrj-HA and Orb2A with DnaJ1-HA with anti Orb2 antibody. Probing the blot with an anti-HA antibody shows the presence of Mrj in the lane for lysate and Orb2 IP suggesting its interaction with Orb2A. In contrast for DnaJ1, it is detected only in the lysate lane and not in the Orb2 IP lane, suggesting no interaction between DnaJ1 and Orb2A. (**F**) The same blot as in E was probed with anti Orb2 antibody and here Orb2A could be detected in both the lysate lanes and Orb2 IP lanes confirming the pull down of Orb2A with anti Orb2 antibody. The data underlying this figure are available at: https://figshare.com/s/f5d913a0a289339ee16b.
(TIFF)

**S1 Data. ClustalO analysis of all the *Drosophila* Mrj isoforms.**
(DOCX)

**S2 Data. Human homologs of other DnaJ interactors of Orb2.**
(DOCX)

**S1 Raw Images. Original, uncropped, and minimally adjusted images of all blots and gels reported in the article.**
(PDF)

## Acknowledgments

We thank Simon Alberti and Randall Halfmann for the yeast strains and the vectors for the chimeric Sup35 prion experiment and the yeast destination vector was a gift from Susan Lindquist's lab. We acknowledge Kausik Si for the Orb2, Konrad Basler for pUASgHA attB, Debashis Mitra for the human DnaJB6, Aravind Penmatsa for the pFastBac-GFP-His, Troy Littleton for the HttQ138, and Norbert Perrimon for the Htt Exon1-GFP constructs used in this paper. Gateway destination plasmids from Drosophila Genomics Resource Center (NIH grant

2P40OD010949), monoclonal antibodies from the Developmental Studies Hybridoma Bank, created by the NICHD of the NIH and maintained at The University of Iowa, and fly stocks from the Bloomington Drosophila Stock Center (NIH P40OD018537) were used in this study. We thank Rishov Goswami, Vishal Naik, Harshada Gadade, Maitheli Sarkar, and Vighnesh Ghatpande for being associated during the initial phases of this work and George Fernandes for helping in making the fly food, stock, and cell line maintenance. We also acknowledge Rahul Bankar, animal house, NCCS for his help in generating the antibodies here and Deepa Subramanyam, Vasudevan Seshadri, Gunther Hollopeter, Nilanjana Das, and Gaurav Agarwal for reading the manuscript and providing valuable suggestions. Schematics in Figs 1A, 1C, 1G, 2A, 2C, 2E, 4F, 4L, 4N, 5B, 5E, 5H, 6A and 7H were created using Biorender.com. Schematics in Figs 1E and 3A were created by hand using Affinity designer software. Full figures were assembled using Affinity designer. AM thanks the spirit of DFHAN for inspiration and dedicates this work to the memory of late KSK.

## Author Contributions

**Conceptualization:** Amitabha Majumdar.

**Data curation:** Meghal Desai, Hemant, Prathamesh Dhamale.

**Formal analysis:** Meghal Desai, Hemant, Ankita Deo, Prathamesh Dhamale.

**Funding acquisition:** Tania Bose, Amitabha Majumdar.

**Investigation:** Meghal Desai, Hemant, Ankita Deo, Jagyanseni Naik, Prathamesh Dhamale, Avinash Kshirsagar, Tania Bose, Amitabha Majumdar.

**Methodology:** Meghal Desai, Hemant, Ankita Deo, Prathamesh Dhamale, Tania Bose, Amitabha Majumdar.

**Project administration:** Tania Bose, Amitabha Majumdar.

**Supervision:** Tania Bose, Amitabha Majumdar.

**Validation:** Meghal Desai, Hemant, Ankita Deo, Prathamesh Dhamale.

**Writing – original draft:** Amitabha Majumdar.

**Writing – review & editing:** Meghal Desai, Hemant, Ankita Deo, Prathamesh Dhamale, Tania Bose, Amitabha Majumdar.

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
