## [Editor Report · Decision Letter 0]

6 Apr 2023

Dear Dr Majumdar, 

Thank you for submitting your manuscript from Review Commons entitled "Mrj an Hsp40 family chaperone regulates the oligomerization of Orb2 and long-term memory" for consideration as a Research Article by PLOS Biology. Please accept my apologies for the delay in getting back to you as we consulted with an academic editor about your submission.

Your revised manuscript and rebuttal has now been evaluated by the PLOS Biology editorial staff, as well as by an academic editor with relevant expertise, and I am writing to let you know that we would like to send your submission for re-review by the original reviewers at Review Commons.

However, before we can send your manuscript back to the reviewers, we need you to complete your submission by providing the metadata that is required for full assessment. To this end, please login to Editorial Manager where you will find the paper in the 'Submissions Needing Revisions' folder on your homepage. Please click 'Revise Submission' from the Action Links and complete all additional questions in the submission questionnaire.

Once your full submission is complete, your paper will undergo a series of checks in preparation for peer review. After your manuscript has passed the checks it will be sent out for review. To provide the metadata for your submission, please Login to Editorial Manager (https://www.editorialmanager.com/pbiology) within two working days, i.e. by Apr 08 2023 11:59PM.

Kind regards,

Richard

Richard Hodge, PhD

Associate Editor, PLOS Biology

rhodge@plos.org

PLOS

---

## [Decision Letter · Decision Letter 1]

11 May 2023

Dear Dr Majumdar,

Thank you for your patience while we considered your revised manuscript "Mrj an Hsp40 family chaperone regulates the oligomerization of Orb2 and long-term memory" for publication as a Research Article at PLOS Biology. Please accept my apologies for the delays that you have experienced during the peer review process. Your revised study has been evaluated by the PLOS Biology editors, the Academic Editor and the original reviewers at Review Commons. Reviewer #1 has provided their review as a point-by-point response to their original comments, which you can also find in the document attached to this letter.

In light of the reviews, which you will find at the end of this email, we would like to invite you to revise the work to thoroughly address the reviewers' reports.

As you will see, the reviewers continue to find your revised version interesting and note that many of their previous comments have been satisfactorily addressed. However, they raise overlapping concerns with the overall strength of the evidence supporting the conclusions made from the behavioral assays. After discussions with the Academic Editor, we feel that including the control data highlighted by the reviewers would strengthen the findings and experimentally addressing the concerns of Reviewer #3 with the behavioral controls and genetic background would be required to continue to consider your manuscript for publication. This would include demonstrating that the phenotypes of the Mrj KO can be rescued by overexpression and to control for the genetic background.

Given the extent of revision needed, we cannot make a decision about publication until we have seen the revised manuscript and your response to the reviewers' comments. Your revised manuscript is likely to be sent for further evaluation by all or a subset of the reviewers.

At this stage, your manuscript remains formally under active consideration at our journal. Given that this represents a large amount of work, we would understand if you would prefer to withdraw their manuscript and pursue faster publication elsewhere. Please notify us by email if you do not intend to submit a revision so that we may withdraw it.

**IMPORTANT - SUBMITTING YOUR REVISION**

*Re-submission Checklist*

*Published Peer Review*

*PLOS Data Policy*

*Blot and Gel Data Policy*

Sincerely,

Richard

Richard Hodge, PhD

Associate Editor, PLOS Biology

rhodge@plos.org

REVIEWS:

Reviewer #1: The authors have send in a revision in which they mostly adequately dealt with my comments (see below, with my reply in red).

Figure 1 (plus related Supplemental figures):

• There seem to be two isoforms of Mrj (like what has been found for human DNAJB6). I find it striking to see that only (preferentially?) the shorter isoform interacts with Orb2. For DNAJB6, the long isoform is mainly related to an NLS and the presumed substrate binding is identical for both isoforms. If this is true for Dm-Mrj too, the authors could actually use this to demonstrate the specificity of their IPs where Orb2 is exclusively non-nuclear?

“Well addressed.”

• I would be interested to know a bit more about the other 5 JDPs that are interactors with Orb2: are the human orthologs of those known? It is striking that these other 5 JDPs interact with Orb2 in Dm (in IPs) but have no impact on Sup35 prion behavior. Importantly, this does not imply they may not have impact on the prion-like behavior of other Dm substrates, including Dm-Orb2.

• “Well addressed. I would like the authors to add the table as supplement to the paper as it is valuable information for follow-up studies. I actually agree that it goes beyond the scope of this paper to show what the biological relevance is of the interaction between these other JDPs and Dm-Orb2.”

• The data in panels H,I indeed suggest that Mrj1 alters the (size of) the oligomers. It would be important to know what is the actual physicochemical change that is occurring here. The observed species are insoluble in 0.1 % TX100 but soluble in 0.1% SDS, which suggest they could be gels, but not real amyloids such as formed by the polyQ proteins that require much higher SDS concentrations (~2%) to be solubilized. This is relevant as Mrj1 reduces polyQ amyloidogenesis whereas is here is shown to enhance Orb2A oligomerization/gelidification. In the same context, it is striking to see that without Mrj the amount of Orb2A seems drastically reduced and I wonder whether this might be due to the fact that in the absence of Mrj a part of Orb2A is not recovered/solubilized due to its conversion for a gel to a solid/amyloid state? In other words: Mrj1 may not promote the prion state, but prevents that state to become an irreversible, non-functional amyloid?

“I am still a bit confused, also because of the complexity of the answer in relation to my question. I do understand and accept that the authors cannot pinpoint (at this stage) what actual physicochemical states of the Orb2A is in under the various condition. What I still do not understand is why there is less Orb2A in the absence of Mrj1. What I also still think could be the case is that Mrj1 is not driving oligomerization per se, but that its (early) interactions with Orb2A affect the oligomerization process (up until becoming amyloids). So, I suggest that the authors remain somewhat open-minded about this alternative explanation and mention it in their paper.”

Figure 2 (plus related Supplemental figures):

• It may be good for clarity to refer to the human Mrj as DNAJB6 according to the HUGO nomenclature. Also, the first evidence for its oligomerization was by Hageman et al 2010.

“Well addressed.”

• It is striking to see that Mrj co-IPs with Hsp70AA, Hsp70-4 but not Hsp70Cb. The fact that interactions were detected without using crosslinking is also striking given the reported transient nature of J-domain-Hsp70 interactions Together, this may even suggest that Mrj-1 is recognized as a Hsp70 substrate (for Hsp70AA, Hsp70-4 but not Hsp70Cb) rather than as a co-chaperone. In fact, a variant of Mrj-1 with a mutation in the HPD motif should be used to exclude this option.

“I understand and am aware of the fact that in some cases JDP-Hsp70 interactions are seen in co-IP even without crosslinking, but also here these may be indirect via binding to substrates for which different Hsp70 may have different specificities (rather than Mrj-1 specifically binding to specific Hsp70 isoforms). In any case, I feel an experiment with an HDP motif mutant would be essential in my view to make any claims on J-domains showing specific binding to only a subset of Hsp70 isoforms.”

• The rest of these data reconfirm nicely that Mrj/DNAJB6 can suppress polyQ-Htt aggregation. Yet note that in this case the oligomers that enter the agarose gel are smaller, not bigger. This argues that Mrj is not an enhancer of oligomerization, but rather an inhibitor of the conversion of oligomers to a more amyloid like state.

“I was indeed wrong regarding my interpretation of figure 2I (I assume incorrectly that the lane marked with Mrj was the knockout). However, the interpretation still is (even more so now) skewed by the fact that the total amount of polyQ detected in the case of cells with Mrj is so much less that in the controls. So, whereas the other anti-aggregation data for polyQ are still solid and confirming literature data, I still feel one cannot interpret that data as to suggest that Mrj is an “enhancer of oligomerization” (for neither polyQ, nor for Orb2A: see above). The situation is likely more subtle.”

Figure 3:

• The finding that knockout of DNAJB6 in mice is embryonic lethal is related to a problem with placental development and not embryonic development (Hunter et al, 1999; Watson et al, 2007, 2009, 2011) as well recognized by the authors. Therefore, the finding that deletion of Dm-Mrj has no developmental phenotype in Drosophila may not be that surprising.

“Well addressed.”

• It is a bit more surprising that Mrj knockout flies showed no aggregation phenotype or muscle phenotype, especially knowing that DNAJB6 mutations are linked to human diseases associated with aggregation (again well recognized by the authors). However, most of these diseases are late-onset and the phenotype may require stress to be revealed. So, while important to this MS in terms of not being a confounder for the memory test, I would like to ask the authors to add a note of caution that their data do not exclude that loss of Mrj activity still may cause a protein aggregation-related disease phenotype. Yet, I also do think that for the main message of this MS, it is not required to further test this experimentally.

“Well addressed.”

Figure 4:

• IPs were done with Orb2A as bite and clearly pulled down substantial amounts of GFP-tagged Mrj. For interactions with Orb2B, a V5-tagged Mrj was use and only a minor fraction was pulled down. Why were two different Mrj constructs used for Arb2A and Orb2b?

“Well addressed.”

• In addition, I think it would be important what one would see when pulling on Mrj1, especially under non-denaturing conditions and what is the status of the Orb2 that is than found to be associated with Mrj (monomeric, oligomeric and what size).

“Well addressed: nice additional data!.”

• This also relates to my remark at figure 1 and the subsequent fractionation experiments they show here in which there is a slight (not very convincing) increase in the ratio of TX100-soluble and insoluble (0.1% SDS soluble) material. My question would be if there is a remaining fraction of 0.1% insoluble (2% soluble) Orb2 and how Mrj affects that? As stated before, this is (only) mechanistically relevant to understanding why there is less oligomers of Orb2 in terms of Mrj either promoting it or by preventing it to transfer from a gel to a solid state. The link to the memory data remains intriguing, irrespective of what is going on (but also on those data I do have several comments: see below).

“Well addressed (still see comments above).”

• I also find the sentence that “Mrj is probably regulating the oligomerization of endogenous Orb2 in the brain” somewhat an overstatement. I would rather prefer to say that the data show that Mrj1 affects the oligomeric behavior/status of Orb2.

“Well addressed.”

Figure 5:

• To my knowledge, the Elav driver regulates expression in all neurons, but not in glial cells that comprise a significant part of the fly heads/brain. The complete absence of Mrj in the fly-heads when using this driver is therefore somewhat surprising. Or do we need to conclude from this that glial cells normally already lack Mrj expression?

“Well addressed.”

• Why not use these lines also for the memory test for confirmation? I understand the concerns of putative confounding effects of a full knockdown (which were however not reported), but now data rely only on the mushroom body-specific knockdown for the 201Y Gal4 line, for which the knockdown efficiency is not provided. But even more so, here a temperature shift (22oC-30oC) was required to activate the expression of the siRNA. For the effects of this shift alone no controls were provided. The functional memory data are nice and consistent with the hypothesis, but can it be excluded that the temperature shift (rather than the Mrj) knockdown has caused the memory defects? I think it is crucial to include the proper controls or use a different knockdown approach that does not require temperature shifts or even use the knockout flies.

“Well addressed. Nice new, consistent data!”

Figure 6:

The finding of a co-IP between Rpl18 and Mrj (one-directional only) by no means suffices to conclude that Mrj may interact with nascent Orb2 chains here (which would be the relevant finding here). The fact that Mrj is a self-oligomerising protein (also in vitro, so irrespective of ribosomal associations!), and hence is found in all fractions in a sucrose gradient, also is not a very strong case for its specific interaction with polysomes. The finding that there is just more self-oligomerizing Orb2A co-sedimenting with polysomes in sucrose gradients neither is evidence for a direct effect of Mrj enhancing association of Orb2A with the translating ribosomes even though it would fit the hypothesis. So all in all, I think the data in this figure and non-conclusive and the related conclusions should be deleted.

“Well addressed with nice additional data.”

Overall, provided that proper controls/additional data can be provided for the key experiments of memory consolidation, I find this an intriguing study that would point towards a role of a molecular chaperone in controlling memory functions via regulating the oligomeric status of a prion-like protein and that is worthwhile publishing in a good journal.

However, in terms of mechanistical interpretations, several points have to be reconsidered (see remarks on figure 1,4); this pertains especially to what is discussed on page 13. In addition, I’d like the authors to put their data into the perspective of the findings that in differentiated neurons DNAJB6 levels actually decline, not incline (Thiruvalluvan et al, 2020), which would be counterintuitive if these proteins are playing a role as suggested here in memory consolidation.

“See my remaining comments above. I think a slight rephrasing of a few of the interpretations would suffice for me to advise acceptation of this nice MS.”

Reviewer #2: The authors have addressed all my concerns about the addition of data (to figure 2G, supplemental figures 5A, B and supplemental figure 6C) and expansion of discussion about Mrj's chaperone functions. 

Two minor points:

- It was mentioned that the Mrj knockout line was created in the w[1118] background. In Fig 6B, is w[1118] the wild type control? Similarly, it appears from the Methods section that the Mrj RNAi and the GFP RNAi lines have the same genetic background. If these are correct, it would be good to point out that the genetic background of the lines being compared in these experiments are equivalent.

- As the other reviewers have pointed out, it would have been stronger to demonstrate that the phenotypes of the Mrj knockout line can be rescued by overexpression. A discussion around why this was not possible would be good to include in the paper.

Reviewer #3: This is a second version of this manuscript which identifies a chaperone of the Hsp40 family (Mrj) that binds Orb2 and modulates its oligomerization, which is critical for Orb2 function in learning and memory in Drosophila. The article starts with a screen of Hsp40 and Hsp70-family proteins that bind Orb2 and the identification of Mrj as the one with the specific interaction. Following the generation of a KO Mrj mutant, they show that the silencing of Mrj in the mushroom body gamma neurons results in weaker memories in a courtship paradigm. The data is novel, interesting, consistent, and generally supportive of the hypothesis. The article also has some concerns as presented. This is an ambitious article that needs additional work before publication.

Several experiments still lack the expected rigor, including controls and quantification. The observations are quite interesting and expansive. As the authors attempt to cover so much ground with Orb2, Htt-103Q, biochem, cell bio, and behavior, the experiments are conducted without the necessary rigor, leaving questions unanswered along with concerns for the interpretation of data. The strength is clearly in the biochemistry with other approaches leaving extensive room for criticism and interpretation.

Importantly, key controls are still missing in the behavioral assays. These assays are very sensitive to genetic background. Multiple controls are necessary to support the claim that an effect on memory is detected both in the Mrj KO experiments and in the RNAi assays. Mrj overexpression is also critical to support the claims throughout the manuscript that Mrj promotes Orb2 oligomerization, which should strengthen memory formation. In addition, no explanation or mechanism are given for why only long-term memory is affected in the Mrj mutant background. These are time-consuming experiments and hard to replicate but controls are critically needed to support the conclusions of the study.

Detailed comments:

I am still not convinced about the ability of Mrj to eliminate Htt-103Q nuclear aggregates. The authors point to literature supporting this anti-aggregation effect of chaperones, but there is significant literature supporting a pro-aggregation effect of chaperones, possibly as a defensive mechanism. This may be an artifact observed in non-physiological models that are comparable to the model shown here: Warrick, Bonini, Nat Gen 1999-No change in polyQ aggregation with Hsp70 co-expression; Fernandez-Funez, Botass, Nature 2000-more compact polyQ aggregates with DNAJ1 co-expression. Other later reports.

Fig 2E show multiple aggregates to show the colocalization of Orb2A and Mrj. Figs 2F and G show drastically decreased Htt aggregates. 2E, F and G don't seem to correlate too well. Counting the number of aggregates per cell could help with this specific point. Also, the middle panel of F still shows a large NI. 

On a more practical point, how does Mrj Lower Htt-109Q aggregation on its own? Hsp40 can bind client proteins but the catalytic activity is provided by Hsp70. Overexpressing Hsp40 can bind substrates and alter their dynamics but should not be sufficient on its own to remove Htt-103Q aggregates

2H: The WB shows similar expression but no loading control. I am curious, what are the different bands in 2H, particularly the one running at 250?

2I: shows Lower oligomers in the presence of Mrj but Total Htt is much lower. Where is the rest of the protein? This is not quantified either

To complete this part of the story, it would be nice if the authors showed the consequence of Mrj KO on Htt-103Q. 

Prion-like terminology: Sorry to make a large point out of this detail. if a protein has a prion-like domain (per the authors), it cannot "convert to a prion-like state". This terminology is misleading even if it can be found in the Orb2 field. Prion-like properties refer to the ability to transition from soluble to misfold and oligomerize, properties that are encoded in the protein, meaning that a soluble state still has prion-like properties. Since the authors make significant references to prions and amyloids, the proper use of the terminology would be desirable. There is an overuse of the "prion" terminology, perhaps to increase the impact of the paper, when the actual intent is to identify aggregated conformations. Later on, Htt aggregates are called oligomers, but Htt does not seem to have a prion-like state. Consistent nomenclature can help avoid distractions and it takes nothing from the merits of the paper.

End of page 4: "… Orb2A's prion-like conversion". Should be conformational conversion

Fig 1J and K: what are the two bands seen in the membranes incubated with anti-Mrj? The second band has a different MW in J and K, it is unclear if this is an Mrj byproduct (fragment?)

Supp fig. 5: Mrj expression / distribution: it would be nice to see the description of Mrj distribution more aligned with the discussion of the Mrj KO line: looking for abnormal phenotypes in brain and muscle makes sense if Mrj is expressed in these tissues.

Fig. 4A-D: These images need controls and quantifications. All images shown have coexpression of Orb2A and Mrj. It would be nice to see controls, particularly of Orb2A and B alone. Is the distribution of Orb2A and / or Orb2B altered by Mrj? Is Mrj contributing to the aggregation of Orb2B since it does not seem to aggregate on its own?

Fig 4B: it seems that the aggregation pattern of the Orb2A325 mutant has changed compared to WT to the formation of fewer but larger aggregates. If this is a consistent change, it is quite relevant. This mutant should be shown alone to determine its aggregation qualities. 

Page 10: Mrj interacts with both the Orb2A and Orb2B isoforms, and this interaction is independent of the RNA binding domain." Technically, the authors only know this for Orb2A and this is relevant because Orb2A and B have different behaviors. 

Fig. 4L: the text describing this panel may need some clarification. "…the presence of both monomeric Orb2B and oligomeric Orb2 in the immunoprecipitated…". I am not sure how the authors distinguish between monomeric Orb2B and oligomeric Orb2 (?) isoforms in the gel after probing with the Orb2 antibody. This may be a typo and refer only to oligomeric vs monomeric Orb2. But since the top of the middle paragraph starts with both Orb2A and B isoforms, this last sentence is confusing.

Fig. 5A: "the detectable form is Orb2B representing its relatively higher abundance in the brain". How do they know this? Are the two isoforms differentiated in WB by electrophoretic mobility?

Fig 5C-D: Now we see here 2 bands that we do not see in Fig 5A. The quantification in 5A makes the point that there is less insoluble Orb2 in the -/-. It would be critical to quantify separately the soluble and the insoluble fractions. If there is less insoluble Orb2, there should be more soluble. Calculating the fraction only indicates that one of them changes but we don't know about the other. This is quite critical in these experiments because this serves as an internal control, just like showing the Tub does not accumulate in the insoluble fraction, which shows good technique in the separation. Since the p = 0.038, quite borderline significance, understanding the differences in each fraction separately is critical.

Figure 6: 6A controls still missing. The background of the Mrj-/- is different from the WT and this can affect the behavioral assay. Isogenic background, adding a line with the same background as the -/- but with normal Mrj, or adding more controls, like a +/- are necessary here.

Same for 6F, more controls are needed, included non-temp shifted and other RNAi lines to account for genetic backgrounds. The mechanistic claims in the paper requires the overexpression of Mrj to promote aggregation and detect stronger memory formation.

Mrj seems to affect Long-term memory but not short-term memory: there is no mechanism for this observation. Without a clear mechanism, it may be just variability or background effects at a time when memories are much more labile. 

Minor:

Fig. 5I is not quantified

Capitalized words (sometimes): prions, yeast, adenine

Please, fix the scale bars to make them legible

---

## [Decision Letter · Decision Letter 2]

29 Feb 2024

Dear Amitabha,

Following on from my previous message, thank you again for your patience while we reconsidered our decision regarding your manuscript entitled "Mrj an Hsp40 family chaperone regulates the oligomerization of Orb2 and long-term memory".

Based on our Academic Editor's assessment of your rebuttal, I am pleased to say that we are likely to accept this manuscript for publication, but we think it would be beneficial to include some of your responses to Reviewer #3 in the manuscript text where appropriate. Specifically, we ask that you please include brief clarifications that were provided in response to points 3 and 4, as well citing and contextualizing the previous work noted in response to points 5 and 6 to lend support for your claims. 

In addition, please also make sure to address the following data and other policy-related requests that I have provided below (A-F):

(A) We would like to suggest the following modification to the title:

"Mrj is a chaperone of the Hsp40 family that regulates Orb2 oligomerization and long-term memory in Drosophila”

(B) You may be aware of the PLOS Data Policy, which requires that all data be made available without restriction: http://journals.plos.org/plosbiology/s/data-availability. For more information, please also see this editorial: http://dx.doi.org/10.1371/journal.pbio.1001797

-Supplementary files (e.g., excel). Please ensure that all data files are uploaded as 'Supporting Information' and are invariably referred to (in the manuscript, figure legends, and the Description field when uploading your files) using the following format verbatim: S1 Data, S2 Data, etc. Multiple panels of a single or even several figures can be included as multiple sheets in one excel file that is saved using exactly the following convention: S1_Data.xlsx (using an underscore).

-Deposition in a publicly available repository. Please also provide the accession code or a reviewer link so that we may view your data before publication. 

Figure 1I, 2I, 2K, 2M, 3H-I, 3K, 5D, 5G, 5J, 5L, 6B, 6F, 7E, S4G, S5C, S6A-B

(C) Please also ensure that each of the relevant figure legends in your manuscript include information on *WHERE THE UNDERLYING DATA CAN BE FOUND*, and ensure your supplemental data file/s has a legend.

(D) We require the original, uncropped and minimally adjusted images supporting all blot and gel results reported in the following Figures:

Figure 1B, 1D, 1H, 1J-K, 2B, 2D, 2F, 2J, 2L, 3B-C, 3J, 4G-K, 4M, 4O-P, 5A, 5C, 5F, 5I, 5K, 6D, 7A-C, 7F-G, S3, S4B, S4E, S5H, S6C-E, S6G, S7A-F

We will require these files before a manuscript can be accepted so please prepare and upload them now. Please carefully read our guidelines for how to prepare and upload this data: https://journals.plos.org/plosbiology/s/figures#loc-blot-and-gel-reporting-requirements

(E) Please ensure that your Data Statement in the submission system accurately describes where your data can be found and is in final format, as it will be published as written there. 

(F) Please also provide a blurb which (if accepted) will be included in our weekly and monthly Electronic Table of Contents, sent out to readers of PLOS Biology, and may be used to promote your article in social media. The blurb should be about 30-40 words long and is subject to editorial changes. It should, without exaggeration, entice people to read your manuscript. It should not be redundant with the title and should not contain acronyms or abbreviations. For examples, view our author guidelines: https://journals.plos.org/plosbiology/s/revising-your-manuscript#loc-blurb

We expect to receive your revised manuscript within two weeks. 

*Published Peer Review History*

*Press*

Sincerely,

Richard

Richard Hodge, PhD

rhodge@plos.org

PLOS

---

## [Editor Report · Decision Letter 3]

12 Mar 2024

Dear Amitabha,

On behalf of my colleagues and the Academic Editor, Josh Dubnau, I am pleased to say that we can in principle accept your manuscript for publication, provided you address any remaining formatting and reporting issues. These will be detailed in an email you should receive within 2-3 business days from our colleagues in the journal operations team; no action is required from you until then. Please note that we will not be able to formally accept your manuscript and schedule it for publication until you have completed any requested changes.

PRESS

Kind regards, 

Richard

Richard Hodge, PhD

rhodge@plos.org

PLOS
